# Direct and indirect pathways for heterosynaptic interaction underlying developmental synapse elimination in the mouse cerebellum
Hisako Nakayama[1,2,3], Taisuke Miyazaki[4], Manabu Abe[5,6], Maya Yamazaki[5], Yoshinobu Kawamura[1,3], Myeongjeong Choo[1], Kohtarou Konno[7], Shinya Kawata[1], Naofumi Uesaka [1], Kouichi Hashimoto[3], Mariko Miyata[2], Kenji Sakimura[5], Masahiko Watanabe[7] & Masanobu Kano [1,8,9] ✉

Developmental synapse elimination is crucial for shaping mature neural circuits. In the neonatal mouse cerebellum, Purkinje cells (PCs) receive excitatory synaptic inputs from multiple climbing fibers (CFs) and synapses from all but one CF are eliminated by around postnatal day 20. Heterosynaptic interaction between CFs and parallel fibers (PFs), the axons of cerebellar granule cells (GCs) forming excitatory synapses onto PCs and molecular layer interneurons (MLIs), is crucial for CF synapse elimination. However, mechanisms for this heterosynaptic interaction are largely unknown. Here we show that deletion of AMPA-type glutamate receptor functions in GCs impairs CF synapse elimination mediated by metabotropic glutamate receptor 1 (mGlu1) signaling in PCs. Furthermore, CF synapse elimination is impaired by deleting NMDA-type glutamate receptors from MLIs. We propose that PF activity is crucial for CF synapse elimination by directly activating mGlu1 in PCs and indirectly enhancing the inhibition of PCs through activating NMDA receptors in MLIs.

Precise formation of neural circuits during development is a prerequisite for proper brain functions[1–3]. In many animal species, diverse and redundant synaptic connections are formed initially, and they are refined subsequently, by strengthening some connections and eliminating others. This process is known as synapse elimination and is widely thought to be crucial for shaping mature neural circuits depending on neural activity [3,4]. Postnatal development of climbing fiber (CF) to Purkinje cell (PC) synapses in the rodent cerebellum has been a representative model of synapse pruning[5–9]. In neonatal mice, each PC receives excitatory synaptic inputs on the soma from more than five CFs originating from neurons in the contralateral inferior olive of the medulla oblongata. The strengths of the multiple CF inputs are similar around birth but during the first postnatal week, inputs from a single CF selectively become stronger than those from the other CFs (functional differentiation)[8,10]. Then, from around postnatal day 9 (P9), only the

strongest CF extends its innervation along growing dendrites of PCs (CF translocation)[8,11]. In parallel, inputs from the weaker CFs on the PC soma are eliminated through two distinct processes from around P7 to around P11 (early phase of CF elimination) and from around P12 to around P17 (late phase of CF elimination)[8,12]. The late phase but not the early phase is known to be dependent on the formation of excitatory synapses on PC distal dendrites from parallel fibers (PFs)[12] that are bifurcated axons of granule cells (GCs) in the cerebellar cortex[13–15]. Through these four developmental processes, most PCs become innervated by single strong CFs on their proximal dendrites by the end of the third postnatal week[8].

Previous studies indicate that neural activity is crucial for CF synapse elimination[8]. Activation of P/Q-type voltage-dependent calcium channel (P/Q-VDCC) in PCs and the resultant calcium signaling is crucial for the selection of a single winner CF, the dendritic translocation of the winner CF,

[1]Department of Neurophysiology, Graduate School of Medicine, The University of Tokyo, Tokyo, Japan. [2]Division of Neurophysiology, Department of Physiology, School of Medicine, Tokyo Women's Medical University, Tokyo, Japan. [3]Department of Neurophysiology, Graduate School of Biomedical and Health Sciences, Hiroshima University, Hiroshima, Japan. [4]Department of Functioning and Disability, Faculty of Health Sciences, Hokkaido University, Sapporo, Japan. [5]Department of Cellular Neurobiology, Brain Research Institute, Niigata University, Niigata, Japan. [6]Department of Animal Model Development, Brain Research Institute, Niigata University, Niigata, Japan. [7]Department of Anatomy, Hokkaido University Graduate School of Medicine, Sapporo, Japan. [8]International Research Center for Neurointelligence (WPI-IRCN), The University of Tokyo Institutes for Advanced Study (UTIAS), Tokyo, Japan. [9]Advanced Comprehensive Research Organization (ACRO), Teikyo University, Tokyo, Japan. ✉e-mail: mkano-tky@m.u-tokyo.ac.jp

and both the early and the late phase of CF elimination[8,16–20]. The P/Q-VDCC is the major high-threshold VDCC in PCs[21] and is shown to be activated by CF synaptic inputs[16] that induce large excitatory postsynaptic potentials with multiple spikes[13,22]. In contrast, the metabotropic glutamate receptor 1 (mGlu1) is required for the late phase of CF elimination[8,23–25]. Immunoelectron microscopic studies show that mGlu1 is expressed at the perisynaptic annuli of postsynaptic densities of both PF and CF synapses[26–28]. A previous study suggests that mGlu1 can be activated at CF synapses particularly when glutamate uptake is inhibited[29]. However, mGlu1 is thought to be functional predominantly at PF synapses[30–34] and trigger signaling cascades involving Gαq, PLCβ3/4, and PKCγ in PCs to fuel CF synapse elimination[35–38]. This argument that mGlu1 is activated at PF-PC synapses is based on the results that NMDA-type glutamate receptors (NMDARs) within the cerebellum are required for CF synapse elimination[39] during the third postnatal week[40] although functional NMDARs are lacking in PCs during this developmental stage[40–42]. Because NMDARs are rich in granule cells (GCs)[43,44], NMDARs are thought to be activated by mossy fiber (MF) excitatory inputs to GCs and mediate neural activity along PFs to PCs to heterosynaptically induce CF elimination. However, there has been no direct experimental evidence for this notion.

The present study aimed to elucidate whether and how MF inputs to the cerebellum refine CF to PC synapses through heterosynaptic interaction in PCs during postnatal cerebellar development. We show that MF to GC synaptic transmission mediated by AMPA-type glutamate receptors (AMPARs), not by NMDARs, and resultant PF activity are crucial for the late phase of CF elimination. Our data suggest that this heterosynaptic interaction between PFs and CFs is mediated by two distinct pathways triggered by PF activity, namely direct activation of mGlu1 in PCs and indirect enhancement of GABAergic inhibition of PCs through activating NMDARs in molecular layer interneurons (MLIs).

## Results

### Generation of a mouse model lacking AMPAR-mediated excitatory transmission specifically at MF to GC synapses during postnatal development

To examine whether neural activity along PFs is required for CF synapse elimination, we established a mouse model in which MF to GC excitatory transmission is abolished. For this purpose, we deleted the AMPAR auxiliary subunit TARPγ2[45,46] specifically from GCs during postnatal development. To delete target molecules from cerebellar GCs, we used GluN2C-Cre mice[47] in which the Cre recombinase is expressed in GCs under the control of the promoter of a GC-specific molecule, GluN2C[43,44]. We crossed the TARPγ2-floxed mice[48] with the GluN2C[+/iCre] mice to obtain GC-specific TARPγ2 knockout mice (TARPγ2-GC-KO mice). As reported previously[49], the immunoreactivity of TARPγ2 protein was seen widely throughout the brain in 8-week-old wild-type mice with the most robust expression in the granule cell layer and the molecular layer of the cerebellum (Supplementary Fig. 1a–c). We found that the TARPγ2 protein was specifically deleted in the granule cell layer of the cerebellum while the expression in other brain regions was not affected in 8-week-old mature TARPγ2-GC-KO mice (Supplementary Fig. 1d–f). We then scrutinized when TARPγ2 was deleted from GCs during postnatal development. A recent study shows that in the cerebellar vermis of GluN2C-Cre mouse, Cre-induced recombination was observed only in lobules 8 and 9 at P7 but spread to the other lobules at P15 and persisted at P21[47]. This expression pattern is consistent with GluN2C mRNA in the cerebellum during postnatal development[50]. Reflecting this posterior to the anterior progression of Cre recombinase expression in the developing cerebellum of GluN2C-Cre mouse, TARPγ2 was almost deleted from the granule cell layer of lobules 9 but it was still present in lobules 4/5 at P14 of TARPγ2-GC-KO mice (Supplementary Fig. 1g–j). As expected, the AMPA receptor subunit GluA2 exhibited the same temporal and spatial expression patterns (Fig. 1a–d).

We examined whether and how AMPAR-mediated membrane currents in GCs were affected in TARPγ2-GC-KO mice. We made whole-cell recordings from GCs in lobules 1-4/5 and in lobules 8-9 of cerebellar slices

from control and TARPγ2-GC-KO mice and recorded AMPAR-mediated membrane currents. We adopted leak-subtracted voltage ramps (from a holding potential of + 40 mV to − 100 mV at a rate of 70 mV/s) to construct instantaneous I–V relationships of the current induced by bath-applied (RS)-AMPA (10 μM). At P17–P19, the amplitudes of AMPAR-mediated currents in TARPγ2-GC-KO GCs tended to be smaller than those in control mice in lobules 8-9 (Fig. 1e, df = 1, F = 3.762, P = 0.079, n = 7 for control and n = 6 for TARPγ2-GC KO mice, Two-way Repeated-Measures ANOVA), whereas they were similar between the two mouse strains in lobules 1-4/5 (Fig. 1f, df = 1, F = 0.143, P = 0.713, n = 6 for both strains of mice, Two-way Repeated-Measures ANOVA). In contrast, when we examined AMPAR-mediated currents in GCs from P60 to P80, they were almost undetectable in TARPγ2-GC-KO mice and were much smaller than in control mice both in lobules 8/9 (Supplementary Fig. 2a, df = 1, F = 24.427, P < 0.001, n = 7 for control and n = 8 for TARPγ2-GC-KO mice, Two-way Repeated-Measures ANOVA) and in lobules 1-4/5 (Supplementary Fig. 2b, df = 1, F = 17.280, P < 0.001, n = 7 for control and n = 8 for TARPγ2-GC-KO mice, Two-way Repeated-Measures ANOVA). These results suggest that AMPAR-mediated excitatory inputs to GCs from MFs tended to be attenuated in lobules 8-9 but they were intact in lobules 1-4/5 in TARPγ2-GC-KO mice at around P17 and that they were almost eliminated in the entire cerebellum of adult TARPγ2-GC-KO mice.

To estimate how the deletion of TARPγ2 from GCs affected their activities in the intact cerebellum of TARPγ2-GC-KO mice, we conducted in vivo whole-cell patch-clamp recordings from GCs in lobules 8-9 during P17–P19 under isoflurane anesthesia. We found that most GCs in control mice exhibited putative spontaneous EPSCs under voltage clamp mode (Fig. 1g, i), and more than 60% of GCs generated spontaneous action potentials under current clamp mode (Fig. 1g, j, k). In contrast, GCs in TARPγ2-GC-KO mice rarely showed putative spontaneous EPSCs (Fig. 1h, i. Control: 4.62 ± 1.40 Hz, n = 22; TARPγ2-GC-KO: 0.05 ± 0.04 Hz, n = 15; P < 0.001, Mann-Whitney U test) and nearly 70% of GCs did not generate sAPs (Fig. 1h, k). The mean sAP frequency in GCs of TARPγ2-GC-KO mice was lower than control mice (Fig. 1j. Control: 0.87 ± 0.42 Hz, n = 22; TARPγ2-GC-KO: 0.53 ± 0.52 Hz, n = 13; P = 0.037, Mann-Whitney U test). Taken together with the data from cerebellar slices (Fig. 1e, f) and from intact cerebella in vivo (Fig. 1g–k), MF to GC excitatory synaptic transmission was almost abolished in lobules 8-9 of TARPγ2-GC-KO mice during P17–P19.

We then examined the spontaneous activities of PCs in lobules 8-9 of the intact cerebellum by in vivo whole-cell recordings under current clamp mode (Supplementary Fig. 3). PCs exhibit two types of spontaneous APs, namely simple spikes and complex spikes. While PCs generate simple spikes depending on the membrane potential, GC activity influences PC's simple spike activity through excitatory synaptic transmission from parallel fibers (PFs), the axons of GCs, onto PCs. In contrast, CF inputs induce very large EPSPs in PC's proximal dendrites and generate characteristic complex spikes[13,17,22]. We found that the simple spike frequency tended to be lower in TARPγ2-GC-KO mice although the difference did not reach the statistically significant level (Supplementary Fig. 3a-c. Control: 35.8 ± 5.6 Hz, n = 10; TARPγ2-KO: 20.3 ± 5.2 Hz, n = 12; df = 20, t = 2.006, P = 0.059, t test). In contrast, the complex spike frequency was higher in TARPγ2-GC-KO mice than in control mice (Supplementary Fig. 3a, b, d. Control: 0.69 ± 0.09 Hz, n = 10; TARPγ2-KO: 1.02 ± 0.11 Hz, n = 12; df = 20, t = -2.257, P = 0.035, t test). The increase in the complex spike frequency might reflect persistent multiple CF innervation of PCs in lobules 8-9 of the adult cerebellum of TARPγ2-GC-KO mice as shown below.

### Impaired CF synapse elimination in cerebellar lobules 8-9 of TARPγ2-GC-KO mice

We then examined whether greatly attenuated MF to GC excitatory transmission and reduced GC activity affected developmental CF synapse elimination in PCs. We made whole-cell recordings from PCs and recorded CF-mediated EPSCs (CF-EPSCs) in lobules 8-9 and 1-4/5 of cerebellar slices from control and TARPγ2-GC-KO mice. To stimulate CFs innervating the

**Fig. 1 | Deficient AMPAR-mediated EPSCs in GCs in lobules 8-9 but not in lobules 1-4/5 of TARPγ2-GC KO mice during the late phase of CF elimination. a–d** Immunohistochemistry for the AMPA receptor subunit GluA2 in lobule 9 (**a** and **b**) and lobule 4/5 (**c** and **d**) of the cerebellum from a control (**a** and **c**) and a TARPγ2-GC-KO (**b** and **d**) mouse at P14. Scale bar, 20 μm. **e, f** Instantaneous I–V relationships of the AMPA (10 μM)-induced current evoked in GCs in lobules 8-9 (**e**) and lobules 1-4/5 (**f**) of the cerebellum from control (white symbols with bold black line, n = 7 for **e** and n = 6 for **f**) and TARPγ2-GC KO (red symbols with red bold line, n = 6 for both **e** and **f**) mice at P17–P19. Data from individual cells are shown with faint lines. Data are mean ± SEM. **g, h** Sample traces of in vivo whole-cell recordings from GCs in lobules 8/9 of a control mouse at P13 (**g**) and a TARPγ2-GC KO mouse at P12 (**h**). Membrane potentials (upper traces) were recorded under the current clamp mode at a resting membrane potential of − 50 mV. Membrane currents (lower traces) were recorded under voltage-clamp mode at a holding potential of − 60 mV. The Inset trace in (**g**) represents an average of spontaneous EPSCs. Scale bars, 40 mV and 5 s (upper traces), 5 pA and 100 ms (lower traces), 2 pA and 10 ms (inset). **i–k** Total absence of spontaneous EPSCs and slightly reduced spontaneous firing frequency in GCs in vivo sampled in lobules 8/9 of TARPγ2-GC KO mice. Summary bar graphs showing the mean frequency of spontaneous EPSCs (**i**), the mean frequency of spontaneous action potentials (**j**), and the frequency distribution of GCs in terms of spontaneous firing frequency (**k**) in control and TARPγ2-GC KO mice at P11–P18. *P < 0.05, ***P < 0.001, Mann-Whitney U test. Data in I and j are mean ± SEM.

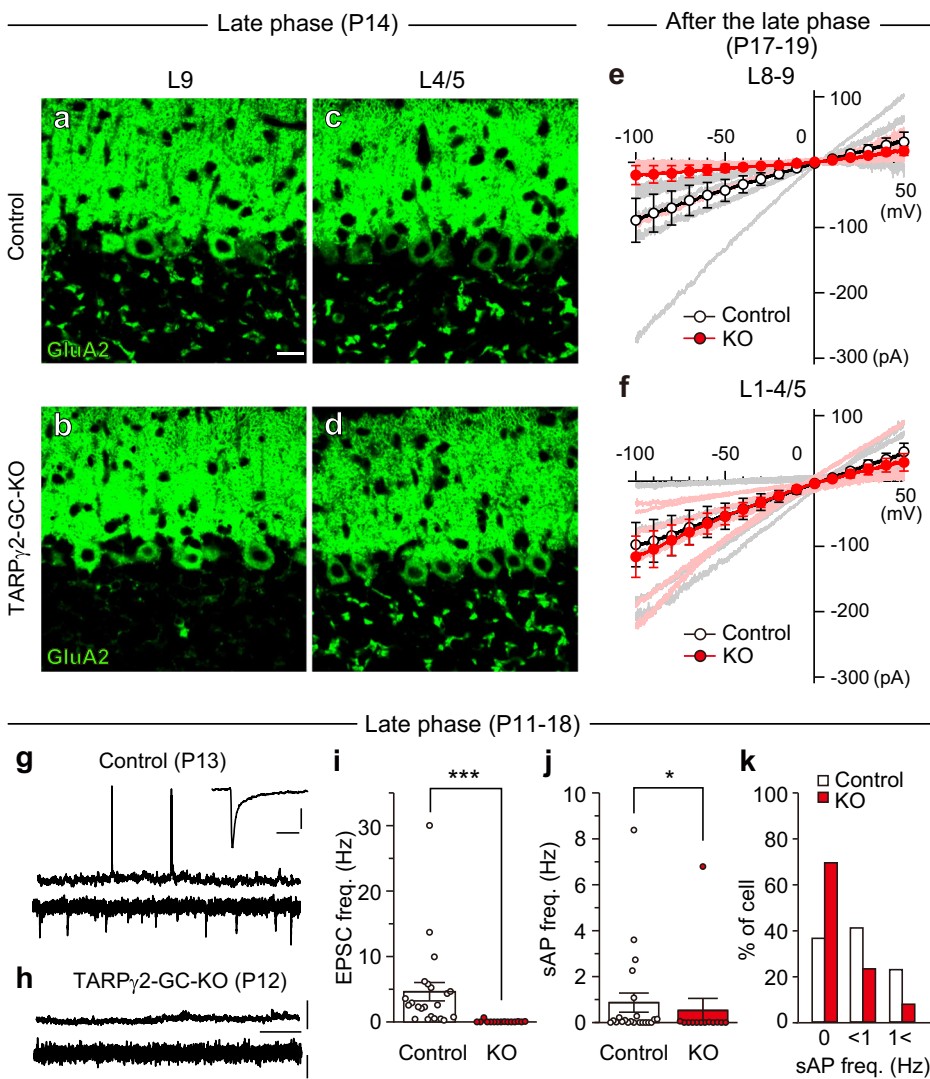

PC under recording, a patch pipette filled with the external solution was placed in the granule cell layer underneath the recorded PC. Under physiological recording conditions, CF stimulation is known to induce EPSCs that show a clear depression to the 2nd of the stimulus pair (paired-pulse depression)[41,42], whereas the stimulation of ascending axons of GCs is reported to induce a clear facilitation (paired-pulse facilitation)[51]. Therefore, we discarded EPSCs that did not exhibit clear paired-pulse depression to avoid possible contamination of EPSCs elicited by unintentionally stimulating GC ascending axons in the GC layer. To search all CFs innervating the recorded PC, we moved the stimulation pipette systematically around the PC soma and increased the stimulus intensity gradually at each stimulation site from 0 to 100 μA[10,18,52]. We estimated the number of CFs innervating the recorded PC from the number of discrete CF-EPSC steps[10,18,52]. During P21–P40 (Adolescent), more than 80% of PCs in control mice were innervated by single CFs in both lobules 8-9 and 1-4/5 (Fig. 2a–d). In contrast, about 60% of PCs were innervated by two or more CFs in lobules 8-9 of TARPγ2-GC-KO mice during P21–P40 (Fig. 2a, b. Control: n = 49; TARPγ2-GC-KO, n = 58; P < 0.0001, Mann-Whitney U test), indicating that CF synapse elimination was impaired during postnatal development until P21. However, the extent of multiple CF innervation was not significantly different between the two mouse strains in lobules 1-4/5 during P21–P40 (Fig. 2c, d. Control: n = 38; TARPγ2-GC-KO, n = 53; P = 0.097, Mann-Whitney U test), suggesting that CF synapse elimination occurred almost normally during postnatal development until P21.

To examine CF innervation patterns morphologically, we labeled subsets of CFs by injecting the anterograde tracer dextran Alexa488 (DA488) into the inferior olive and performed triple labeling for calbindin (a PC marker), VGluT2 (a CF terminal marker), and DA488. As exemplified in Fig. 2e, f1, f2, DA488-labeled CFs followed proximal dendrites of PCs, and their axon terminals were completely overlapped with VGluT2 immunoreactivity, indicating that such PCs were innervated by single CFs. By contrast, PCs of TARPγ2-GC-KO mice were often associated with DA488/VGluT2-double positive CF terminals together with DA488-negative/VGluT2-positive CF terminals on their somata and the bottom of proximal dendrites (Fig. 2g, h1, h2), indicating that such PCs were innervated by at least two CFs with different cellular origins in the inferior olive. In lobules 4/5, DA488-labeled CFs and their axon terminals were completely overlapped with VGluT2 immunoreactivity in both controls (Fig. 2i, j1, j2) and TARPγ2-GC-KO (Fig. 2k, l1, l2) mice, showing typical mono CF innervation patterns.

We then examined the extent of CF translocation along PC dendrites and the degree of persistent CF innervation on the PC soma. We found that the relative height of VGluT2-positive CF terminals in lobules 8-9 was higher in 8-week-old adult TARPγ2-GC-KO mice than in age-matched control mice (Fig. 3a1, a2, b1, b2, e. Control: 67.9 ± 0.61%, n = 282 (i.e., the number of distal tips of VGluT2-positive CF terminals); TARPγ2-GC-KO: 70.9 ± 0.62%, n = 264; P < 0.001, Mann-Whitney U test) but the height in lobules 1-4/5 was similar between the two mouse strains (Fig. 3c1, c2, d1, d2,

**Article**

**Fig. 2 | Persistent multiple CF innervation of PCs in lobules 8-9 but not in lobules 1-4/5 of adolescent TARPγ2-GC-KO mice. a–d** Sample CF-ESPC traces (**a**, **c**) and frequency distribution histograms of PCs in terms of the number of discrete CF-EPSC steps (**b**, **d**) for control (open columns) and TARPγ2-GC-KO (filled columns) mice aged P21–P40. Holding potentials for (**a**, **c**), − 10 mV. Scale bars for (**a**, **c**), 10 ms, and 0.5 nA. ***P < 0.001, Mann-Whitney *U* test. **e−l** Triple immunofluorescence labeling for the PC marker calbindin (blue or gray), anterogradely-labeled (DA488-positive) CFs (red), and the CF terminal marker VGluT2 (green) in lobule 8 (**e−h**) and lobule 4/5 (**i−l**) of a control (**e**, **f1**, **f2**, **i**, **j1**, **j2**) and a TARPγ2-GC-KO (**g**, **h1**, **h2**, **k**, **l1**, **l2**) mouse at 3 weeks of age. Boxed regions in **e**, **g**, **i**, and **k** are enlarged in **f1** and **f2**, **h1** and **h2**, **j1** and **j2**, and **l1** and **l2**, respectively. Arrows in **h1** and **h2** indicate anterogradely unlabeled (DA488-negative) VGluT2-positive CF terminals. Scale bars: **e**, 20 μm; **f1**, 5 μm.

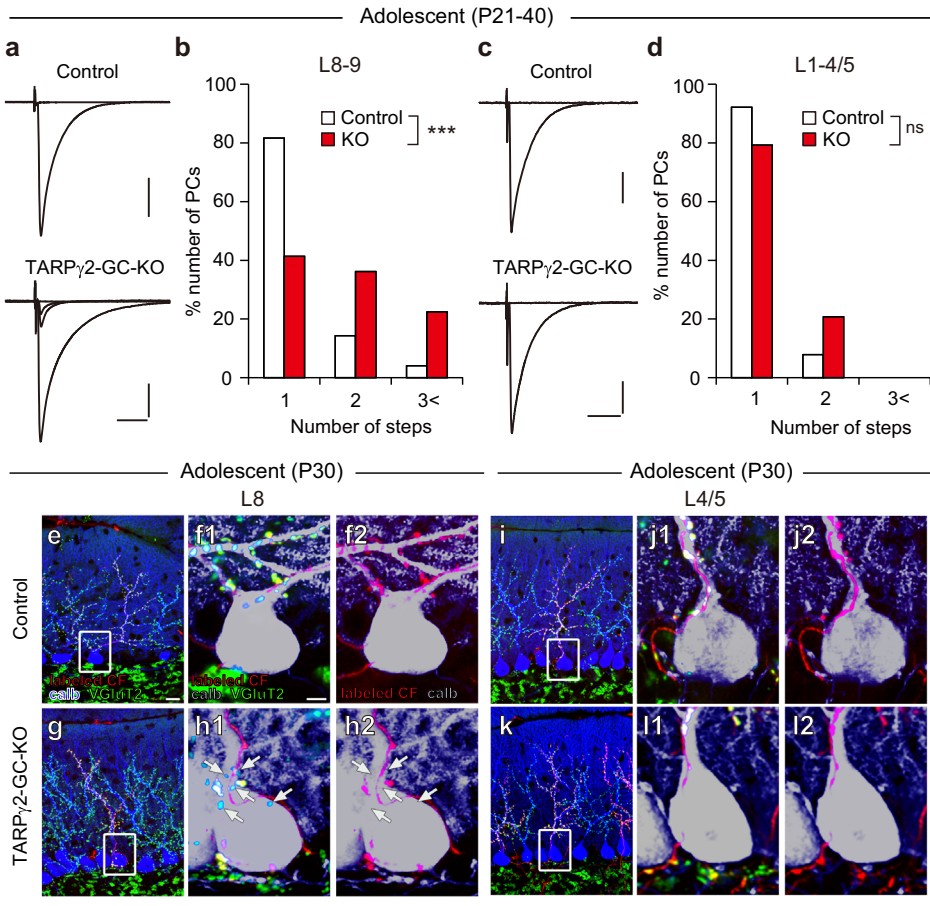

e. Control: 75.3 ± 0.39%, *n* = 333; TARPγ2-GC-KO: 76.3 ± 0.39%, *n* = 348; *P* = 0.069, Mann-Whitney *U* test). The density of VGluT2-positive CF terminals on the PC soma was higher in 8-week-old adult TARPγ2-GC-KO mice than in age-matched control mice in both lobules 8-9 (Fig. 3a1, b1, f. Control: 0.30 ± 0.02, *n* = 120; TARPγ2-GC-KO: 0.51 ± 0.04, *n* = 126; *P* < 0.001, Mann-Whitney *U* test) and lobules 1-4/5 (Fig. 3c1, d1, f. Control: 0.13 ± 0.01, *n* = 132; TARPγ2-GC-KO: 0.18 ± 0.02, *n* = 137; *P* = 0.013, Mann-Whitney *U* test), but the difference was much more prominent in lobules 8-9. These morphological observations indicate that in TARPγ2-GC-KO mice, the translocation of CFs to PC dendrites was enhanced and the elimination of CF synapses from the PC soma was impaired in lobules 8-9. Besides, the elimination of CF synapses from the PC soma was impaired in lobules 1-4/5, although less prominently than in lobules 8-9, suggesting that CF synapse elimination was slightly impaired in the anterior lobules of TARPγ2-GC-KO mice.

Since the elimination of redundant CF synapses and extension of CF innervation along PC dendrites are greatly affected by PF-PC synapse formation[12,20,53,54], we examined whether the morphology of PF-PC synapses was altered in TARPγ2-GC-KO mice. Immunohistochemistry targeting VGluT1, a marker for PF terminals, revealed punctate signals in the molecular layer of lobules 9 and 4/5 in both mouse models (Supplementary Fig. 4a, c, e, g). Electron microscopy analysis (Supplementary Fig. 4b, d, f, h) showed that each PC spine (asterisks) made contact with a single PF terminal in lobules 9 and 4/5 in both mice. Quantitative analysis demonstrated that the density of PF synapses was similar between the two mouse strains in both lobules 9 and 4/5 (Supplementary Fig. 4i. PF synapse density per 100 μm². Lobule 9, Control: 29.5 ± 1.5, *n* = 24 regions from 3 mice; TARPγ2-GC-KO: 29.9 ± 0.9, *n* = 24 regions from 3 mice; *P* = 0.3279, Mann-Whitney *U* test. Lobule 4/5, Control: 24.2 ± 1.2, *n* = 24 regions from 3 mice, TARPγ2-GC-KO: 24.4 ± 1.6, *n* = 24 regions from 3 mice, *P* = 0.8302, Mann-Whitney *U* test). Moreover, the size of PF terminals was not different

between the two genotypes in lobules 9 and 4/5 (Supplementary Fig. 4j. Lobule 9, Control: 0.21 ± 0.009, *n* = 108 terminals from 3 mice, TARPγ2-GC-KO: 0.21 ± 0.01, *n* = 109 terminals from 3 mice, *P* = 0.8737, Mann-Whitney *U* test. Lobule 4/5, Control: 0.21 ± 0.01, *n* = 87 terminals from 3 mice, TARPγ2-GC-KO: 0.24 ± 0, 0.01, *n* = 87 terminals from 3 mice, *P* = 0.1713, Mann-Whitney *U* test). We then examined whether PF-PC synaptic transmission was altered in TARPγ2-GC-KO mice. We recorded EPSCs elicited by stimulating PFs in the middle of the molecular layer by changing the stimulus intensity. The obtained stimulus-response relationships were similar between the two mouse strains in both lobules 8-9 (Supplementary Fig. 4k, 4m. Control, *n* = 13, TARPγ2-GC-KO, *n* = 14, dF = 1, *F* = 0.362, *P* = 0.553, Two-way Repeated-Measures ANOVA) and 4/5 (Supplementary Fig. 4l, 4n. Control, *n* = 15, TARPγ2-GC-KO, *n* = 14, dF = 1, *F* = 0.00354, *P* = 0.953, Two-way Repeated-Measures ANOVA). These results collectively indicate that PF-PC synapses in both the anterior and posterior lobules are normal in TARPγ2-GC-KO mice both morphologically and electrophysiologically.

**The late phase of CF elimination mediated by mGlu1 signaling in PCs is impaired in cerebellar lobules 8-9 of TARPγ2-GC-KO mice**
We then examined the developmental course of CF innervation of PCs from P6 to P18 in lobules 8-9. The frequency distributions in terms of the number of CFs innervating individual PCs decreased similarly in control and TARPγ2-GC-KO mice from P6 to P15 and there were no differences between the genotypes during P6–P8 (Fig. 4a. Control, *n* = 25; TARPγ2-GC-KO, *n* = 34; *P* = 0.736, Mann-Whitney *U* test), P10–P12 (Fig. 4b. Control, *n* = 44; TARPγ2-GC-KO, *n* = 46; *P* = 0.807, Mann-Whitney *U* test) and P13–P15 (Fig. 4c. Control, *n* = 44; TARPγ2-GC-KO, *n* = 29; *P* = 0.241, Mann-Whitney *U* test). However, during P16–P18, the percentage of PCs innervated by multiple CFs was higher in TARPγ2-GC-KO mice than in control mice (Fig. 4d. Control, *n* = 70; TARPγ2-GC-KO, *n* = 63,

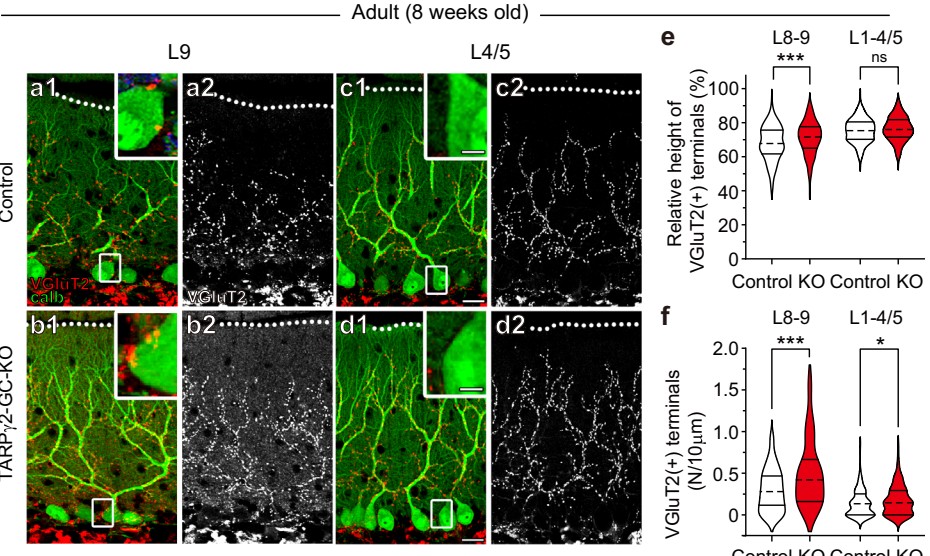

**Fig. 3 | Enhanced CF translocation and impaired elimination of somatic CF synapses in lobules 8-9 but not in lobules 1-4/5 of adult TARPγ2-GC-KO mice. a–d** Double immunostaining for VGluT2 (red) and calbindin (green) (**a1, b1, c1, d1**) and single immunostaining of VGluT2 (**a2, b2, c2, d2**) in lobule 9 (**a, b**) and lobule 4/5 (**c, d**) from a control (**a, c**) and a TARPγ2-GC-KO (**b, d**) mouse at 8 weeks of age. Dotted lines indicate the pial surface. Boxed regions in (**a1, b1, c1, d1**) are enlarged as insets in individual figure panels. Scale bars in (**c1**) and (**d1**), 20 μm for the lower magnification images and 5 μm for the higher magnification images. **e, f** Summary violin plots for the relative height of the tip of VGluT2-positive CF terminals (**e**), representing the shortest distance from the middle of the PC layer to the distal tip of the VGluT2-positive puncta divided by the molecular layer thickness. $n = 282$ and 333 in lobules 8-9 and1-4/5, respectively, from 3 control mice; $n = 264$ and 348 in lobules 8-9 and1-4/5, respectively, from 3 TARPγ2-GC-KO mice) and the number of VGluT2-positive CF terminals around the PC soma (**f**, representing the number of VGluT2-positive CF terminals per 10 μm length of the PC somatic membrane. $n = 120$ and 132 in lobules 8-9 and 1-4/5, respectively, from 3 control mice; $n = 126$ and 137 in lobules 8-9 and1-4/5, respectively, from 3 TARPγ2-GC-KO mice) in lobules 8-9 and lobules 1-4/5 from control (open plots) and TARPγ2-GC-KO (filled plots) mice at 8 weeks of age. The dashed line within each violin plot represents the median, while the solid lines at the top and bottom indicate the 75th and 25th percentiles, respectively. *$P < 0.05$, ***$P < 0.001$, $t$ test.

$P = 0.015$, Mann-Whitney $U$ test). These results indicate that the initial formation of CF-PC synapses and subsequent elimination of redundant CF synapses until P13–P15 proceed normally, but the late phase of CF elimination is specifically impaired in lobules 8-9 of TARPγ2-GC-KO mice.

To check whether the effect of the impaired CF synapse elimination persists into adulthood, we examined the CF innervation of PCs during P60 to P80. We found that the degrees of multiple CF innervation in lobules 8-9 were higher in TARPγ2-GC-KO mice than in control mice (Fig. 4e. Control, $n = 45$; TARPγ2-GC-KO, $n = 31$; $P < 0.001$, Mann-Whitney $U$ test), which is almost the same as the results for P21–P30 (Supplementary Fig. 5a. Control, $n = 33$; TARPγ2-GC-KO, $n = 49$; $P < 0.001$, Mann-Whitney $U$ test) and for P31–P40 (Supplementary Fig. 5c. Control, $n = 16$; TARPγ2-GC-KO, $n = 19$; $P < 0.001$, Mann-Whitney $U$ test). These results indicate that the impaired CF synapse elimination during the third postnatal week was not corrected subsequently, and the aberrant CF innervation persisted into adulthood in TARPγ2-GC-KO mice. In contrast, CF innervation of PCs was normal in lobules 1-4/5 of TARPγ2-GC-KO mice during P21-P30 (Supplementary Fig. 5b. Control, $n = 19$; TARPγ2-GC-KO, $n = 23$; $P = 0.443$, Mann-Whitney $U$ test), P31–P40 (Supplementary Fig. 5d. Control, $n = 19$; TARPγ2-GC-KO, $n = 30$; $P = 0.181$, Mann-Whitney $U$ test) and P60–P80 (Fig. 4f. Control, $n = 32$; TARPγ2-GC-KO, $n = 29$; $P = 0.081$, Mann-Whitney $U$ test) although AMPAR-mediated currents in GCs were almost deleted in lobules 1-4/5 during P60–P80 (Supplementary Fig. 2b). These results suggest that MF to GC excitatory transmission is crucial for CF synapse elimination during the third postnatal week, but it is dispensable for maintaining normal CF innervation thereafter. Thus, the third postnatal week is thought to be the critical period for CF synapse elimination that depends on MF to GC excitatory transmission.

Our previous studies show that mGlu1 is involved in the late phase of CF elimination during the third postnatal week[8,24]. We, therefore, examined whether the impaired CF synapse elimination in TARPγ2-GC-KO mice involved mGlu1 signaling. We performed RNAi-mediated knockdown of

mGlu1 in PCs by injecting a lentivirus carrying microRNA against mGlu1 and cDNA for EGFP under the control of the PC-specific L7 promoter into lobules 8-9 of the cerebellum of control and TARPγ2-GC-KO mice at P0-P1. We then examined CF innervation in EGFP-positive mGlu1-knockdown PCs and EGFP-negative control PCs in the same cerebellar slices during P21–P37. We found that mGlu1 knockdown in the control mouse cerebellum increased the percentage of PCs with multiple CF innervation (Fig. 4g, h. Control, $n = 32$; mGlu1-PC-KD, $n = 32$; $P < 0.05$, Mann-Whitney $U$ test). In contrast, the degree of multiple CF innervation was not different between PCs with mGlu1 knockdown and those without knockdown in TARPγ2-GC-KO mice (Fig. 4i, j. TARPγ2-GC-KO, $n = 32$; TARPγ2-GC-KO + mGlu1-PC-KD, $n = 32$; $P = 0.752$, Mann-Whitney $U$ test), indicating that the effect of mGlu1 knockdown was occluded in PCs of TARPγ2-GC-KO mice. These results suggest that CF synapse elimination that depends on MF to GC excitatory synaptic transmission requires mGlu1 signaling in PCs during the third postnatal week.

**NMDARs in GCs are dispensable for CF synapse elimination**

Previous studies show that blockade of NMDARs within the cerebellum during postnatal development from P4-P5[39] or P15[40] causes persistent multiple CF innervation, indicating that NMDARs within the cerebellum are required for the late phase of CF elimination. Since functional NMDARs are absent in PCs during the third postnatal week[40–42], NMDARs in cell types other than PCs in the cerebellum should be responsible. Since NMDARs are rich at MF to GC synapses[43,44], NMDARs in GCs were considered to be involved in CF synapse elimination. To test this possibility, we generated mice with GC-specific deletion of the core NMDAR subunit GluN1 (GluN1-GC-KO mouse) by crossing GluN1-floxed mice with GluN2C-Cre mice. Our immunohistochemical data show that GluN1 was richly expressed in the cerebral cortex and the hippocampus in control and GluN1-GC-KO mice (Fig. 5a, d). In the cerebellum, GluN1 immuno-positive puncta were found in the granule cell layer in control mice

**Fig. 4 | Impairment of the late phase of CF elimination mediated by mGlu1 in PCs in cerebellar lobules 8-9 of TARPγ2-GC-KO mice.**
**a–d** Developmental changes in CF innervation. Frequency distribution histograms of PCs in terms of the number of discrete CF-EPSC steps during P6–P8 (**a**), P10–P12 (**b**), P13–P15 (**c**), and P16–P18 (**d**) in cerebellar lobules 8-9 from control (white columns) and TARPγ2-GC-KO (red columns) mice. *$P < 0.05$, Mann-Whitney $U$ test. **e, f** CF innervation in adult PCs. Frequency distribution histograms of PCs in terms of the number of discrete CF-EPSC steps during P60-P80 in cerebellar lobules 8-9 (**e**) and 1-4/5 (**f**) from control (white columns) and TARPγ2-GC-KO (red columns) mice. ***$P < 0.001$, Mann-Whitney $U$ test. **g–j** Occlusion of impaired CF synapse elimination by mGlu1 knockdown in PCs of TARPγ2-GC-KO mice. Sample CF-EPSC traces (**g, i**) and frequency distribution histograms in terms of the number of discrete CF-EPSC steps (**h, j**) for mGlu1 knockdown in PCs of control mice aged P21 to P37 (**g, h**) and TARPγ2-GC-KO aged P22 to P34 (**i, j**). Holding potentials for (**g, i**), − 10 mV. Scale bars for **g, i**, 10 ms, and 0.5 nA. *$P < 0.05$, Mann-Whitney $U$ test.

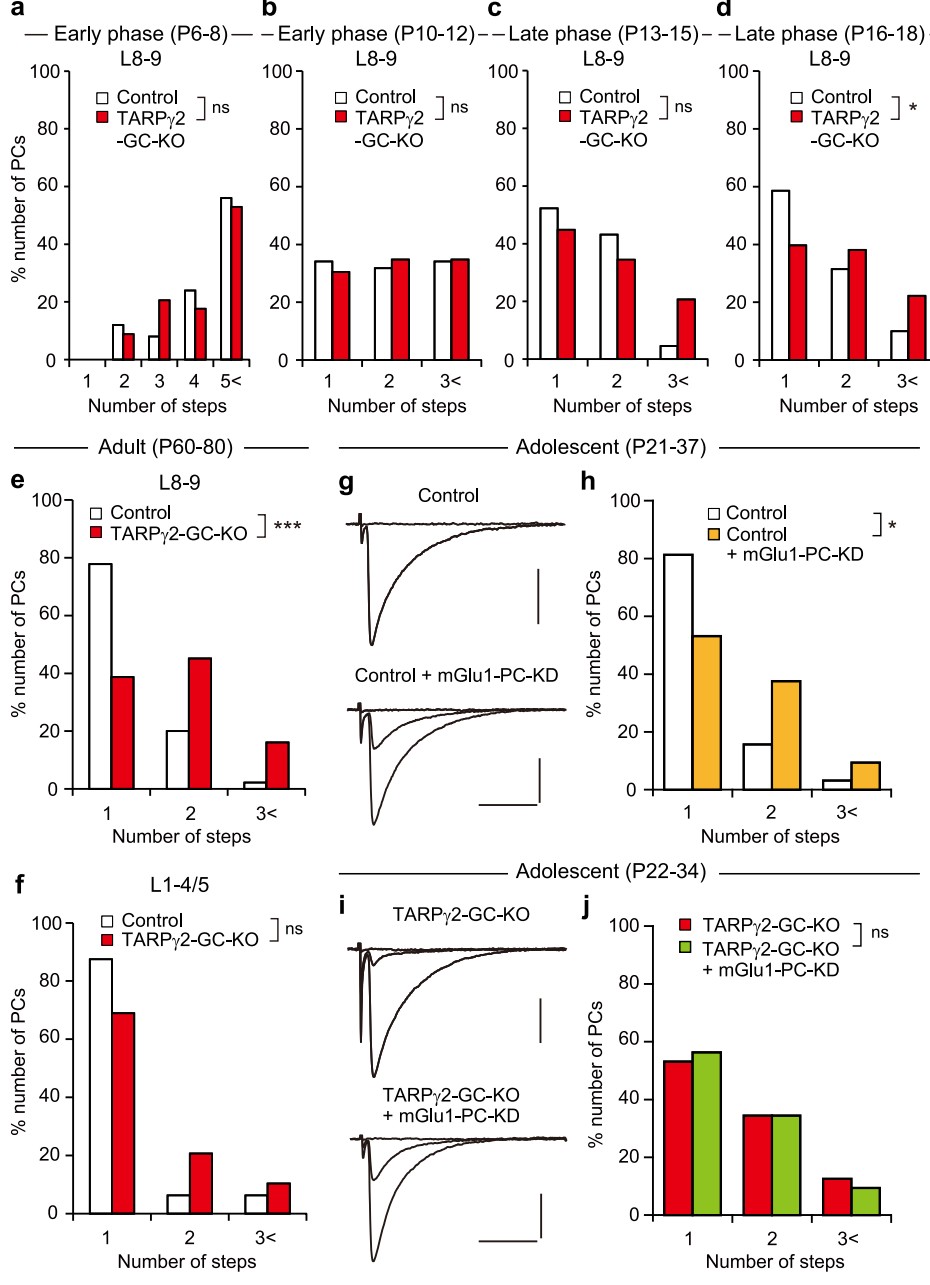

(Fig. 5b, c1, c2) but they were absent in GluN1-GC-KO mice (Fig. 5e, f1, f2), indicating that GluN1 was specifically deleted from GCs in GluN1-GC-KO mice. We then performed whole-cell recordings from GCs in cerebellar slices at P15–P17 and recorded EPSCs evoked by stimulating putative MFs nearby at the holding potential of − 70 mV and + 40 mV. We found that the NMDAR-mediated slow component of EPSC was almost absent in GluN1-GC-KO mice (Fig. 5g, h. Control: 0.46 ± 0.06, $n = 22$; GluN1-GC-KO: 0.08 ± 0.01, $n = 22$; $P < 0.001$, Mann-Whitney $U$ test), indicating that functional NMDA receptors were virtually absent at MF to GC synapses of GluN1-GC-KO mice at P15–P17.

We then examined CF innervation of PCs in lobules 8-9 and in lobules 1-4/5 during P29-P46 but found no significant difference between control and GluN1-GC-KO mice in CF innervation patterns of PCs either in lobules 8-9 (Fig. 5i, j. Control: $n = 25$; GluN1-GC-KO: $n = 23$; $P = 0.951$, Mann-Whitney $U$ test) or in lobules 1-4/5 (Fig. 5k. Control: $n = 26$; GluN1-GC-KO: $n = 22$; $P = 0.939$, Mann-Whitney $U$ test). We also examined CF innervation during P17–P19 but also found no significant difference between the two mouse strains either in lobules 8-9 (Fig. 5l, m. Control:

$n = 29$; GluN1-GC-KO: $n = 25$; $P = 0.282$, Mann-Whitney $U$ test) or in lobules1-4/5 (Fig. 5n. Control: $n = 26$; GluN1-GC-KO: $n = 23$; $P = 0.687$, Mann-Whitney $U$ test). These results suggest that NMDARs in GCs are dispensable for CF synapse elimination. Thus, NMDA receptors in cell types other than GCs should be responsible for CF synapse elimination in the developing cerebellum.

**NMDARs in MLIs are involved in the late phase of CF elimination**
Basket cells and stellate cells, collectively called molecular layer interneurons (MLIs), receive excitatory inputs from PFs and exert GABAergic inhibition to PCs and other MLIs[14]. Stellate cells are shown to express NMDARs in their dendrites such that they surround the postsynaptic density[55]. These extrasynaptic NMDARs in stellate cell dendrites can be activated by high-frequency PF stimulation through glutamate spillover[56–59] and enhance GABAergic inhibition to PCs[57]. We have previously shown that GABAergic inhibition from MLIs to PCs during P10–P16 is required for CF synapse elimination[52]. These results raise a possibility that NMDARs on MLIs are responsible for CF synapse elimination. To address this issue, we generated a

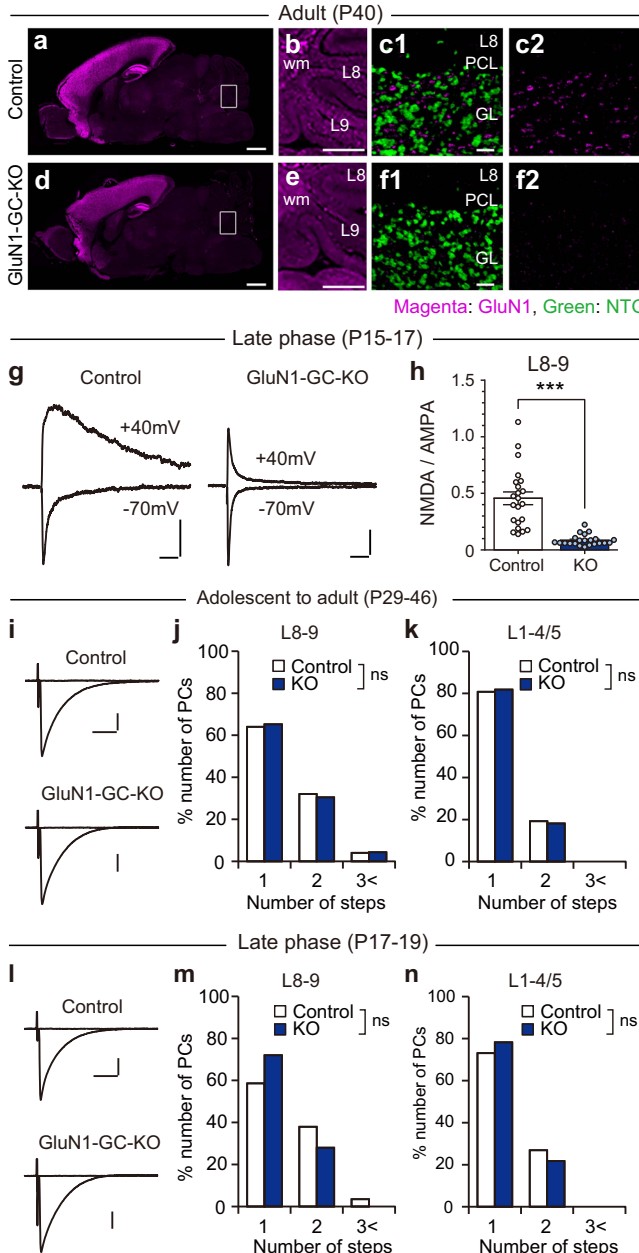

**Fig. 5 | NMDARs in GCs are dispensable for CF synapse elimination.**
**a–f** Immunohistochemistry for the NMDAR subunit GluN1 and Neuro Trace Green (NTG) in a control (**a, b, c1, c2**) and a GluN1-GC-KO (**d, e, f1, f2**) mouse at P40. Boxed regions in (**a**) and (**d**) are enlarged in (**b**) and (**e**), respectively. Scale bar: 1000 μm for (**a**) and (**d**), 500 μm for (**b**) and (**e**), 20 μm for (**c1**) and (**f1**). **g, h** Sample traces (**g**) and a summary bar graph showing the ratio of the NMDAR component over the AMPAR component (**h**) of EPSCs recorded from GCs by stimulating MFs at a holding potential of −70 mV and +40 mV in cerebellar lobules 8-9 of control and GluN1-GC-KO mice. The amplitude of the AMPAR component and that of the NMDAR component were measured at the peak of EPSC with a holding potential of −70 mV and at 20 ms after the stimulus with a holding potential of +40 mV, respectively. ***$P < 0.001$, Mann-Whitney $U$ test. Data in (**h**) are mean ± SEM. **i–n** Normal CF synapse elimination in GluN1-GC-KO mice. Sample CF-EPSC traces from PCs in lobules 8-9 (**i, l**) and frequency distribution histograms in terms of the number of discrete CF-EPSC steps in lobules 8-9 (**j, m**) and lobules 1-4/5 (**k, n**) of control (white columns) and GluN1-GC-KO (dark blue columns) mice aged P29 to P46 (**i–k**) and P17 to P18 (**l–n**).

conditional KO mouse, NR1-MLI/PC-KO mice, in which NMDA receptors in GluD2-expressing cells are deleted by crossing GluN1-floxed mice and GluD2-Cre mice[60]. Since GluD2 and Cre are expressed in both MLIs and PCs but not in the other neurons in the cerebellum of the GluD2-Cre mice[60], GluN1 was expected to be deleted from MLIs and PCs. By conducting SDS-digested freeze-fracture replica labeling of immunogold particles for GluN1 and PSD95, we found that GluN1 particles were found beside PSD95 particles in dendrites of MLIs in control mice (Supplementary Fig. 6a, c). In contrast, GluN1-particles were absent in MLI dendrites of GluN1-MLI/PC-KO mice (Supplementary Fig. 6b, c. Control: 76.1 ± 9.1, $n = 79$ sections from 3 mice; GluN1-MLI/PC-KO: 4.7 ± 1.6, $n = 39$ sections from 3 mice; $P < 0.0001$, Mann-Whitney $U$ test) while PSD95-particles were found similarly to control mice (Supplementary Fig. 6b). We then performed whole-cell recordings from MLIs in cerebellar slices at P15–P17 and applied burst stimulation to PFs (5 pulses at 100 Hz) at holding potential of −70 mV in 0 mM Mg$^{2+}$- and 10 mM glycine-containing bath solution (Supplementary Fig. 6d, e)[57–59]. In MLIs of control mice, fast components of PF-mediated EPSCs were blocked by 10 μM NBQX (Supplementary Fig. 6d left middle panel), and the remaining slow components were abolished by further addition of the NMDAR antagonist R-CPP (5 μM) (Supplementary Fig. 6d, left bottom panel), indicating that the fast and slow components were mediated by AMPARs and NMDARs, respectively. These results confirm that extrasynaptic functional NMDARs are present in MLI dendrites and they can be activated by PF synaptic activity in control mice.

In MLIs of GluN1-MLI/PC-KO mice at P15-P17, NBQX abolished the fast components (Supplementary Fig. 6d right middle panel) and the further addition of R-CPP abolished the remaining slow components (Supplementary Fig. 6d right bottom panel) of PF burst-induced EPSCs similarly to control mice. However, NMDAR-mediated EPSCs tended to be smaller, while AMPAR-mediated EPSCs were similar in peak amplitude and charge transfer to those of control mice (Supplementary Fig. 6f. Peak amplitude of the 1$^{st}$ AMPAR component. Control: 111 ± 19.0 pA, $n = 13$; GluN1-MLI/PC-KO: 140 ± 17.8 pA, $n = 8$; $P = 0.111$, Mann-Whitney $U$ test. Supplementary Fig. 6g. Peak amplitude of NMDAR component. Control: 53.5 ± 13.2 pA, $n = 13$; GluN1-MLI/PC-KO: 24.4 ± 7.07 pA, $n = 8$; $P = 0.060$, Mann-Whitney $U$ test. Supplementary Fig. 6i. Charge transfer of the 1$^{st}$ AMPAR component. Control: 0.27 ± 0.07 pAs, $n = 13$, GluN1-MLI/PC-KO: 0.30 ± 0.04 pAs, $n = 8$, $P = 0.277$, Mann-Whitney $U$ test. Supplementary Fig. 6j. Charge transfer of the NMDAR component. Control: 15.3 ± 4.79 pAs, $n = 13$, GluN1-MLI/PC-KO, 6.16 ± 1.88 pAs, $n = 8$, $P = 0.128$, Mann-Whitney $U$ test). However, the ratio of the amplitude or the charge transfer of the NMDAR-EPSC to that of the 1$^{st}$ AMPAR-EPSC was significantly smaller in GluN1-MLI/PC-KO mice than in control mice (Supplementary Fig. 6h. Amplitude. Control: 0.57 ± 0.12, $n = 13$; GluN1-MLI/PC-KO: 0.19 ± 0.05, $n = 8$; $P = 0.011$, Mann-Whitney $U$ test. Supplementary Fig. 6k. Charge transfer. 67.9 ± 15.6, $n = 13$, GluN1-MLI/PC-KO, 23.7 ± 7.15, $n = 8$, $P = 0.036$, Mann-Whitney $U$ test). These results suggest that the degree of PF-induced activation of NMDARs in MLIs was reduced in GluN1-MLI/PC-KO mice.

In the PCs of control mice, we confirmed that functional NMDA receptors were absent at both CF-PC and PF-PC synapses during the third postnatal week[40–42]. CF-EPSCs and PF-EPSCs in PCs of control mice at P17 were abolished by 10 μM NBQX and further addition of the NMDAR antagonist R-CPP (5 μM) had no effect (Supplementary Fig. 7a1−3, b1−3, e, f). However, in PCs of 7-week-old control mice, both CF-EPSCs and tetanus-induced PF-EPSCs were partially sensitive to R-CPP (Supplementary Fig. 7c1−3, d1−3, e, f). These results confirm previous reports that CF-EPSCs partially contain an NMDAR-mediated component after 5 weeks of age[61,62].

We then examined whether CF synapse elimination was altered in GluN1-MLI/PC-KO mice. We found a modest but significant increase of PCs innervated by two or three CFs in GluN1-MLI/PC KO mice at P30-P50 (Fig. 6a, b. Control: $n = 66$; GluN1-MLI/PC-KO: $n = 68$; $P = 0.022$, Mann-Whitney $U$ test). During postnatal development, there was no significant difference between control and GluN1-MLI/PC KO mice in the frequency

**Fig. 6 | NMDA receptors in MLIs are involved in CF synapse elimination. a, b** Impaired CF synapse elimination in GluN1-MLI/PC-KO mice aged P30 to P50. Sample CF-EPSC traces (**a**) and frequency distribution histograms in terms of the number of discrete CF-EPSC steps (**b**) in control (white columns) and GluN1-MLI/PC-KO (green columns) mice. Holding potential, − 10 mV. Scale bars, 10 ms and 0.5 nA. *P < 0.05, Mann-Whitney *U* test. **c, d** Developmental changes in CF innervation. Frequency distribution histograms of PCs in terms of the number of discrete CF-EPSC steps during P11−P13 (**c**), and P15−P18 (**d**) in control (white columns) and GluN1-MLI/PC-KO (green columns) mice. *P < 0.05, Mann-Whitney *U* test. **e, f** Triple immunofluorescence labeling for calbindin (blue or gray), anterogradely labeled (DA488-positive) CFs (red), and VGluT2 (green) in a control mouse at P30. Scale bar: 20 μm for (**e**), 5 μm for (**f**). **g, h** Triple immunofluorescence labeling similar to (**e**) and (**f**) but for data from a GluN1-MLI/PC-KO mouse. **i, j** Triple immunofluorescence labeling similar to (**g**) and (**h**) but for data from another GluN1-MLI/PC-KO mouse. Arrows in (**h**) and (**j**) indicate anterogradely unlabeled (DA488-negative) VGluT2-positive CF terminals on PCs.

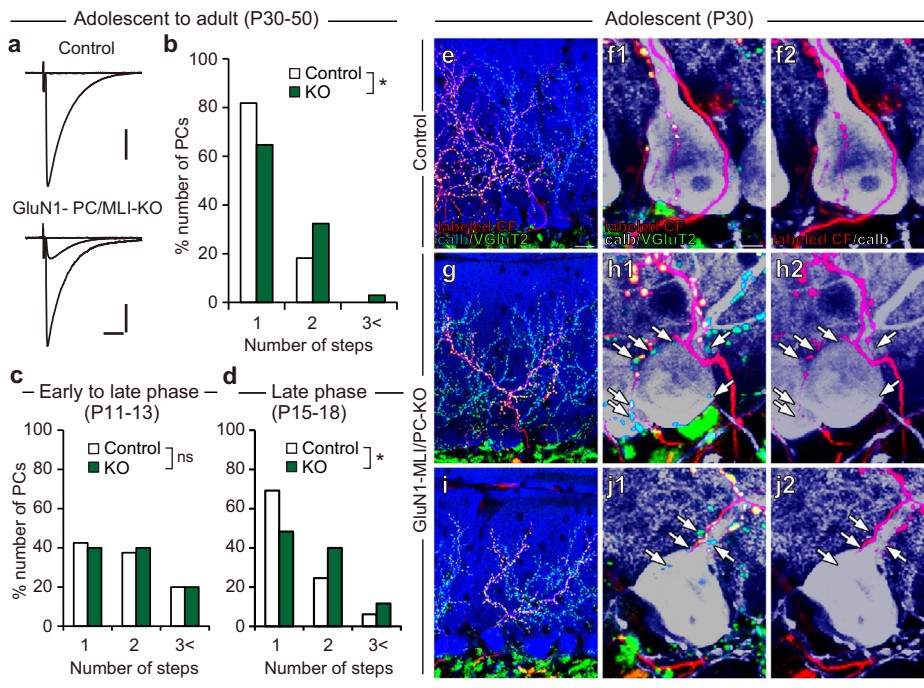

distribution of PCs in terms of the number of CF-EPSC steps during P11−P13 (Fig. 6c. Control: *n* = 40; GluN1-MLI/PC KO: *n* = 30; *P* = 0.883, Mann-Whitney *U* test) but the difference between the genotypes was significant during P15−P18 (Fig. 6d. Control: *n* = 64; GluN1-MLI/PC KO: *n* = 60; *P* = 0.022, Mann-Whitney *U* test). These results suggest that CF synapse elimination during the third postnatal week was specifically impaired in GluN1-MLI/PC KO mice. To examine CF innervation patterns morphologically, we labeled subsets of CFs by injecting the anterograde tracer DA488 into the inferior olive and performed triple labeling for calbindin, VGluT2, and DA488. We found that DA488-labeled CFs were associated with PC proximal dendrites and their axon terminals were overlapped with VGluT2 immunoreactivity in control mice (Fig. 6e, f1, f2). By contrast, PCs of GluN1-MLI/PC KO mice were often associated with DA488/VGluT2-double positive CF terminals together with DA488-negative/VGluT2-positive CF terminals on their somata and proximal dendrites (Fig. 6g, h1, h2, i, j1, j2), indicating that such PCs were innervated by at least two CFs with different cellular origins in the inferior olive. Taken together, these results suggest that NMDARs in MLIs are involved in the late phase of CF synapse elimination.

## Discussion

Previous studies indicate that PFs and CFs compete for postsynaptic sites on PC dendrites during postnatal development and in adulthood. For example, in GluD2 knockout mice, impairment of PF-PC synapse formation appeared during the second postnatal week, the density of PF-PC synapses was reduced to about half that of wild-type mice, and PC dendritic spines without PF synaptic contacts (free spines) are frequently observed[63,64]. In parallel with the impaired PF-PC synapse formation, CFs form ectopic synapses on distal dendrites of neighboring PCs by extending exuberant collaterals, resulting in persistent multiple CF innervation[53,54]. In contrast, in cerebella with retracted CF innervation of PCs such as in rats or mice in which neural activity was suppressed by tetrodotoxin application to the cerebellum[65,66] and in global or PC-selective P/Q-VDCC knockout mice[16,20], PF synaptic territory expanded proximally along PC dendrites and ectopic spines with PF terminals were often generated (hyper-spiny transformation)[65]. Since PF synapse formation occurs massively during the second and third postnatal weeks, its impairment greatly increases the degree of multiple CF innervation with the same time course of the late phase of CF elimination. The occupancy of postsynaptic spines at PC distal dendrites by PFs restricts the CF innervation territory to PC proximal dendrites and provides a basis for the late phase of CF elimination. However, it was not known whether activity along PFs contributes to CF synapse elimination. Our present results from TARPγ2-GC-KO mice unequivocally show that PF activity is indispensable for the late phase of CF elimination. CF synapse elimination was impaired in cerebellar lobules 8-9 in which TARPγ2 was deleted from GCs and AMPAR-mediated currents in GCs were greatly reduced during the third postnatal week. In contrast, CF synapse elimination proceeded almost normally in cerebellar lobules 1-4/5 in which TARPγ2 expression in GCs and AMPAR-mediated currents were not reduced during the third postnatal week. Notably, normal CF mono-innervation was observed in PCs in lobules 1-4/5 in the mature stage when TARPγ2 expression and AMPAR-mediated currents in GCs were almost totally deleted. These results ssuggest that the third postnatal week is the critical period for CF synapse elimination that requires MF-GC-PF activity and that neural activity along this pathway does not contribute to the maintenance of mono CF innervation in the mature cerebellum.

We found that the extent of CF innervation along PC dendrites was larger in lobules 8-9 of TARPγ2-GC-KO mice than in control mice (Fig. 3e), suggesting that CF translocation is enhanced in TARPγ2-GC-KO mice. In contrast, we observed no difference in the morphology and density of PF-PC synapses, the size of PF terminals, or synaptic transmission elicited by PF stimulation between the two mouse strains (Supplementary Fig. 4), suggesting that PF-PC synapses are normal in TARPγ2-GC-KO mice. Net excitation of PCs by PF inputs should be reduced in TARPγ2-GC-KO mice because their GCs exhibited decreased spontaneous activities in vivo (Fig. 1i, j). In contrast, the CF activity and net excitation by CF inputs were elevated in TARPγ2-GC-KO mice because the frequency of complex spikes in vivo was significantly larger in TARPγ2-GC-KO mice than in control mice (Supplementary Fig. 3d). The dominance of CF inputs over PF inputs to excite PCs is thought to underlie the enhanced CF translocation in TARPγ2-GC-KO mice. In contrast, the expansion of CF innervation may reduce PF-PC synaptic territory over PC dendrites, but we found that PF-PC synapses were normal. Since the inactivation of PC activity increases PF innervation territories over PC dendrites[65,66], the reducing effect of PF innervation territories by CF activity might be counter-balanced by the enhancing action of PF innervation due to the reduced PF-induced excitation of PCs in TARPγ2-GC-KO mice.

Previous studies show that mGlu1 in PCs is required for the late phase of CF elimination[23,24,28], but it was not clear whether mGlu1 is activated at PF-PC synapses or at CF-PC synapses to trigger its downstream cascade for CF synapse elimination. Repetitive PF stimulation readily activates postsynaptic mGlu1 and induces slow EPSP/EPSCs[30,32,34] or IP3-mediated calcium release in PC distal dendrites[31,33]. In contrast, activation of mGlu1 at CF synapses was shown to be restricted by strong glutamate uptake by glutamate transporters[29]. It is assumed that mGlu1 is activated by PF inputs and hererosynaptically facilitates the elimination of redundant CF synapses mainly during the third postnatal week[8,40]. In the present study, we showed that CF synapse elimination was impaired in mice with PC-specific mGlu1 knockdown but this effect was occluded in TARPγ2-GC-KO mice (Fig. 4g-j). This result supports the notion that mGlu1 is activated at PF-PC synapses to fuel CF synapse elimination during the third postnatal week. However, mGlu1 signaling in PCs is not considered to be required for the maintenance of CF mono-innervation in the mature cerebellum, since conditional deletion of mGlu1 from PCs in adult mice caused no change in CF innervation[67], which is consistent with the present result of normal CF innervation in lobules 1-4/5 of mature TARPγ2-GC-KO mice (Fig. 2c, d). It is reported that spatially clustered PF burst inputs can cause the accumulation of released glutamate and can activate mGlu1 at PF-PC synapses nearby through the spread of released glutamate[68,69]. Such a PF burst may heterosynaptically activate mGlu1 at CF-PC synapses located near the activated PF bundles, which may explain part of the heterosynaptic interaction between PF- and CF-PC synapses. However, the major event of the mGlu1-mediated late phase of CF elimination is the removal of redundant CF synapses from PC somata[11]. Since somatic CF synapses are spatially distant from PF-PC synapses on PC dendrites, the major effect of PF-to-CF heterosynaptic interaction should be transmitted intracellularly from PC dendrites to somata.

Previous studies indicate that NMDARs in the cerebellum are required for CF synapse elimination[39] particularly during the third postnatal week[40]. Since functional NMDARs are absent at either PF-PC or CF-PC synapses during the second and third postnatal weeks[40–42], NMDARs in other cell types should be responsible for CF synapse elimination. Since NMDARs are richly expressed in GCs during postnatal development[43,44], it has been speculated that NMDARs contribute to MF to GC excitatory transmission. In cerebellar slice preparations, NMDAR-mediated currents were reported to amplify high-frequency MF inputs by prolonging the time courses of synaptic inputs, but NMDARs had no contribution to EPSPs/EPSCs in response to single MF stimulation at 1 Hz[70]. Therefore, NMDARs may contribute to CF synapse elimination by transmitting high-frequency MF inputs to GCs and enhancing PF activity. However, we found normal CF synapse elimination in mice with GC-specific deletion of GluN1 (Fig. 5i–n). In awake mice with P60-P80 of age, GCs were shown to exhibit sparse high-frequency bursts of action potentials during locomotion with high instantaneous firing frequencies of around 100 Hz[71]. However, NMDAR-mediated currents had negligible contributions, if any, to the GC burst activities[71] due to the voltage-dependent Mg$^{2+}$ block of NMDAR channels. Together, these results suggest that NMDARs in GCs are dispensable for CF synapse elimination.

We found that the late phase of CF elimination was impaired in NR1-MLI/PC-KO mice (Fig. 6). Because functional NMDARs are lacking in PCs during the second and third postnatal weeks[40–42], the impaired CF synapse elimination was considered to result from the deletion of NMDARs in MLIs. Since PF burst stimuli can induce NMDAR-mediated currents in MLIs in cerebellar slices[56–59], NMDARs may contribute to the activity of MLIs in vivo particularly when GCs exhibit high-frequency bursts of action potentials.

We found that the degree of impairment of CF synapse elimination in NR1-MLI/PC-KO mice (Fig. 6b) was milder than in TARPγ2-GC-KO mice (Fig. 2b, P < 0.01, Mann-Whitney U test). In contrast, the PC-specific mGlu1 knockdown in control mice (Fig. 4h) caused a similar degree of impairment in CF synapse elimination to that of TARPγ2-GC-KO mice (Fig. 4j, P = 0.916, Mann-Whitney U test). These results suggest that under our experimental conditions, activation of NMDARs in MLIs by PF inputs may play only a supplementary role in CF synapse elimination, whereas the GC activity facilitates CF synapse elimination mainly through activating mGlu1 at PF-PC synapses. The elevation of MLI activity by PF-induced NMDAR activation may contribute only partially to the total spiking activity of MLIs since MLIs exhibit spontaneous firing at 1-35 Hz in vivo[72]. Therefore, the deletion of NMDARs from MLIs may have reduced GABAergic inhibition onto PCs only partially. Moreover, NMDARs may not have been completely deleted in MLIs of NR1-MLI/PC-KO mice, which may have resulted in only a partial reduction of MLI-mediated GABAergic inhibition onto PCs in response to high-frequency PF activity. Indeed, we found a tendency of reduction but not complete elimination of the NMDAR-mediated component of PF-EPSCs in MLIs of NR1-MLI/PC-KO mice (Supplementary Fig. 6). We noticed that among 8 MLIs of NR1-MLI/PC-KO mice, 3 cells had almost no NMDAR-mediated component, whereas 5 cells exhibited clear NMDAR-components comparable to control mice, suggesting that NMDAR-deficient and NMDAR-intact MLIs coexisted in the cerebellum of NR1-MLI/PC-KO mice. Whichever the reason is, reduced GABAergic inhibition to PCs is considered to result in impairment of CF synapse elimination, which is consistent with the previous reports that reduced GABAergic inhibition from basket cells to PCs causes impairment of CF synapse elimination from around P10 to P16[52,73].

Our results suggest that activation of mGlu1 in PCs and NMDARs in MLIs, which resulted from the accumulation of released glutamate in a spatially restricted manner[68,69], is required for the late phase of CF elimination. Such a glutamate accumulation may be caused by clustered PF burst inputs, but there has been no evidence for the presence of such PF bursts in vivo during postnatal development. We observed a low frequency of GC spontaneous firing at P11-18 (Fig. 1g, i) and no obvious mGlu1-mediated slow EPSPs in PCs in vivo at P13-17 (Supplementary Fig. 3). However, our in vivo recordings from GCs and PCs were performed under anesthesia and therefore it is possible that physiological activities of these neurons were masked. Indeed, a previous study showed that GCs exhibited sparse high-frequency burst firings during locomotion in awake mice with P60-P80 of age[71]. If such a high-frequency burst activity is present in clusters of GCs during the developmental stages for CF synapse elimination, mGlu1 in PCs and NMDARs in MLIs can be activated by clustered PF burst inputs. Moreover, patterned spontaneous activity is present in various regions of the developing nervous system, including "retinal waves" that are crucial for shaping mature neural circuits of the visual system[74]. In the developing cerebellum, patterned "traveling waves" of spontaneous firings of PCs are present[75], and CF inputs are highly synchronized during the first postnatal week[76]. It is therefore possible that clusters of GCs may also exhibit patterned synchronized firings and generate clustered PF burst inputs during early postnatal development.

In conclusion, our present results strongly suggest that PF inputs heterosynaptically influence CF synapse elimination in the developing cerebellum. The heterosynaptic interaction in PCs derived from direct activation of mGlu1 at PF-PC synapses and indirect enhancement of GABAergic inhibition onto PCs involving NMDA receptors in MLIs. While mGlu1 has been shown to trigger signaling cascades in PCs involving Gαq[37], PLCβ3 and PLCβ4[36,38], PKCγ[35], Semaphorin 7 A[77], and BDNF[78], it is not well understood how GABAergic inhibition regulates CF synapse elimination. Whether these two pathways converge or act independently in PCs to eliminate redundant CF synapses to establish mature synaptic wiring in the cerebellum remains to be investigated.

## Methods

### Animals

All experiments were approved by the experimental animal ethics committees and the biosafety committee for living modified organisms of The University of Tokyo, Tokyo Women's Medical University, Hiroshima University, Niigata University, and Hokkaido University. We used male and female TARPγ2(flox/flox):GluN2C(cre/wt) mice, GluN1(flox/flox):-GluN2C(cre/cre) mice, and GluN1(flox/flox):GluD2(cre/wt) mice aged from P6 to P80 for the present experiments. They are referred to as

TARPγ2-GC-KO mice, GluN1-GC-KO mice, and GluN1-MLI/PC-KO mice, respectively. Age-matched littermates without cre were used for control experiments. We maintained all mice in conditions on a reversed 12-hour light/dark cycle with free access to food and water.

## Lentiviral injection

As described previously[77,78], we designed VSV-G pseudotyped lentiviral vectors (pCL20c) (Hanawa et al., 2002) for PC-specific expression under the control of a truncated L7 promoter (pCL20c-L7)[79]. We designed the following engineered miRNAs for mGlu1 knockdown by using a BLOCK-iT Pol II miR RNAi expression vector kit (K4935-00, Invitrogen, CA, USA): 5'-TGCTGAAATCAGGGAGTCTCTGATGAGTTTTGGCCACTGACTG ACTCATCAGACTCCCTGATTT-3' and 5'-CCTGA AATCAGGGAGT CTGATGAGTCAGTCAGTGGCCAAAACTCATCAGAGACTCCCTG ATTTC-3'. For virus injection into the cerebellum, a 33-gauge Hamilton syringe filled with a viral solution was attached to a micropump (UltramicroPump II, World Precision Instruments (WPI)). Under anesthesia by inhalation of isoflurane (1–2%), 4–5 µl (4–5 × 10^5 TU) of the viral solution was injected at a rate of 200 nl/min using a microprocessor-based controller (Micro4, WPI) into the cerebellar lobules 8-9 of C57BL/6 N mice at P0–1. The syringe was left for an additional 2 min before it was withdrawn. The scalp was then sutured, and the mice were returned to their home cages.

## In vitro whole-cell patch-clamp recordings

Mice were deeply anesthetized by $CO_2$ or isoflurane inhalation and decapitated. The brains were quickly removed and parasagittal slices with a thickness of 250 µm were prepared from the cerebellar vermis with a vibratome slicer (VT1200S, Leica) in a chilled artificial cerebrospinal fluid (ACSF) containing 125 mM NaCl, 2.5 mM KCl, 2 mM $CaCl_2$, 1 mM $MgSO_4$, 1.25 mM $NaH_2PO_4$, 26 mM $NaHCO_3$, and 20 mM glucose, bubbled with 95% $O_2$ and 5% $CO_2$. Cerebellar slices were kept at 25 °C in the normal ACSF. When we recorded from GCs or MLIs, cerebellar slices were initially incubated in the normal ACSF at 35 °C for 30 min before keeping them at 25 °C. Whole-cell recordings were made from visually identified PCs, GCs, or MLIs using an upright microscope (BX50WI, Olympus, or Axio Examiner, Zeiss). The normal ASCF was used as a bath solution during recordings. Patch pipettes were filled with an intracellular solution composed of 60 mM CsCl, 10 mM CsD-gluconate, 20 mM TEA-Cl, 20 mM BAPTA, 4 mM $MgCl_2$, 4 mM ATP, 0.4 mM GTP, and 30 mM HEPES (pH 7.3, adjusted with CsOH). Bicuculline (10 µM, Tocris) or picrotoxin (100 µM, Tocris) was added to the bath solution to block inhibitory synaptic transmission for recording EPSCs. CFs were stimulated by using a patch pipette filled with the normal ACSF that was placed in the internal granule cell layer around the recorded PC soma. Square electrical pulses with 100 µs duration for CF stimulation were applied at 0.2 Hz. The stimulation pipette was systematically moved around the PC soma. The number of CFs innervating the recorded PC was estimated as the number of discrete EPSC steps during gradually increasing the stimulus intensity at each stimulus location. AMPAR-mediated membrane currents were measured from GCs during voltage ramp from + 40 mV to − 100 mV at a rate of 70 mV/s by subtracting the currents in the control bathing solution from those in the presence of (RS)-AMPA (10 µM). In this experiment, the bathing solution contained bicuculline (10 µM, Tocris), strychnine (10 µM, Sigma), cyclothiazide (100 µM, Tocris), and TTX (1 µM, Nacalai). To evoke PF-mediated EPSCs in MLIs, a stimulation pipette was placed in the molecular layer and five square pulses were applied at 100 Hz. In this experiment, $Mg^{2+}$ ions were omitted and glycine (10 µM), picrotoxin (100 µM), and strychnine (10 µM) were added to the bathing solution. All experiments were performed and the data were analyzed under conditions that the experimenters were blind to mouse genotypes or experimental operations. All data were recorded at 32 °C with an EPC10 patch clamp amplifier with Patch Master software (HEKA Elektronik). Online data acquisition and offline data analysis were performed using Fit Master software (HEKA Elektronik).

## In vivo whole-cell patch-clamp recordings

Animals were anesthetized with isoflurane (2–4%). The depth of anesthesia was monitored by vibrissae movements and withdrawal reflex to hindlimb pinch. Body temperature was maintained at 37 ± 1 °C using a heating pad. The animal's head was fixed on a stereotaxic apparatus (Narishige) and the surface of the cerebellar vermis was exposed as described previously[17]. Whole-cell patch-clamp recordings from GCs and PCs were made with a Multiclamp 700B amplifier (Molecular Devices) by in vivo blind patch-clamp techniques[17]. Patch pipettes were filled with the intracellular solution with the following compositions (in mM): 133 KMeSO_3, 7.4 KCl, 10 HEPES, 3 $Na_2ATP$, 0.3 $Na_2GTP$, 0.3 $MgCl_2$, 0.05–0.1 EGTA, pH 7.3. Liquid-junction potentials were not compensated. Electrophysiological data were filtered at 3 or 10 kHz and digitized at 20 kHz with an ITC-16 interface (Instrutech) and acquired with Axograph X software (Axograph Scientific). Data were analyzed with Axograph X, Excel (Microsoft), and OriginPro (OriginLab). The frequencies of spontaneous action potentials (APs) and EPSCs in PCs and GCs were calculated as the number of events divided by the recording time of 30 seconds when the baseline of membrane current or potential became stable enough after establishing whole-cell recordings.

## Light and electron microscopic analyses

Mice were deeply anesthetized with pentobarbital and perfused transcardially with 4% paraformaldehyde in 0.1 M phosphate buffer (PB, pH 7.4) for light microscopic analysis or 2% paraformaldehyde and 2% glutaraldehyde in 0.1 M PB for electron microscopic analysis. For immunohistochemistry, brains were cut in half at the midline, dehydrated, and embedded in paraffin, or processed into 50 µm cerebellar sections using a vibratome (Leica). The paraffin sections containing age-matched control and TARPγ2-GC-KO mice were immunostained with antibodies against GluA2 (rabbit, 847-863aa; RRID, AB_2571754)[48] and TARPγ2 (rabbit, 302-318aa; RRID, AB_2571844)[48] followed by amplification with Tyramide Signal Amplification (TSA, Perkin Elmer). The vibratome sections were immunostained with goat anti-calbindin (RRID, AB_2532104)[80], guinea pig anti-VGluT2 (RRID, AB_2341096)[81], and guinea pig anti-VGluT1 (RRID, AB_2571618)[81] antibodies. For anterograde labeling of CFs with dextran Alexa488 (DA488, Invitrogen), mice were deeply anesthetized with isoflurane, secured in the stereotaxic frame, and injected with DA488 into the right inferior olive. A needle was inserted into the medulla, and injections were done unilaterally at 1.0 mm rostral to the rostral tip of the occipital bone, 1.5 mm right to the midline, and 1.8 mm ventral to the pial surface, with tilting the manipulator by 47 degrees. The tracer was enhanced immunohistochemically. For electron microscopy, 50 µm cerebellar sections were incubated with 1% $OsO_4$ for 20 min and 2% uranyl acetate for 20 min and dehydrated in graded alcohol and embedded in Epon 812. The ultrathin sections were processed with an ultramicrotome (Leica EM UC7, Leica). The electron micrographs were taken with electron microscopy (JEM1400, JEOL) and analyzed with the Metamorph software (Molecular devices).

## SDS-digested freeze-fracture replica labeling

SDS-digested freeze-fracture replica labeling (SDS-FRL) was performed as described previously[82]. Mice were perfused with 2% PFA in 0.1 M PB (pH 7.2) for 10 min and processed into 140 µm vibratome cerebellar sections. Small regions containing the molecular layer were rapidly frozen by a high-pressure freezing machine (HPM010, Leica Microsystems) and their replicas were processed by a freeze-fracture replica machine (JFD V, JEOL). The replicas were incubated with rabbit anti-GluN1 antibody (RRID, AB_2571604) (1 µg/ml)[83], followed by incubation with species-specific gold particle-conjugated secondary antibody (5 nm in diameter, British Biocell International), guinea-pig anti-PSD95 antibody (RRID, AB_2571612) (1 µg/ml)[84] and species-specific gold particle-conjugated secondary antibody (10 nm in diameter, British Biocell International). Each incubation was done overnight at 15°C. After intensive washing, the replicas were placed onto grids coated with formvar. Images were taken with an electron microscope (JEM 1400, JEOL). Image analysis was conducted using the

MetaMorph software (Molecular Devices). We defined straight, longitudinally fractured regions (less than 1 μm in diameter) containing clustered intramembranous particles as protoplasmic faces (P-face) of MLI dendrites. Within these regions, we defined the area labeled for PSD95, an excitatory postsynaptic marker, and facing the extraplasmic face (E-face) of nerve terminals as the PF-MI postsynaptic region, because PFs are sole excitatory inputs to MLIs.

## Statistical analysis

No data were excluded from the analysis. Throughout the text and figures, n represents the number of cells analyzed unless otherwise stated in the figure legends. Averaged values were represented as mean ± SEM. Statistical significance was assessed by two-sided $t$-test or Mann–Whitney $U$ test, depending on whether the data sets passed the normality test and equal variance test, unless otherwise stated in the text. Statistical analyses were conducted with SigmaPlot 12.1 or 12.5 (Systat Software) and $p$ values smaller than 0.001 were described as $p < 0.001$, otherwise actual $p$ values were described in the text. Differences between the two samples were considered statistically significant if the $p$-value was less than 0.05.

## Reporting summary

Further information on research design is available in the Nature Portfolio Reporting Summary linked to this article.

## Data availability

The datasets generated during and/or analyzed during the current study are available from the corresponding author upon reasonable request. The numerical source data for the figures were provided as Supplementary Data 1.

## Code availability

The analysis code used in the current study is available from the corresponding author upon reasonable request.

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

## Acknowledgements
We deeply thank Takaki Watanabe for helpful discussion and K. Matsuyama and M. Karigane for excellent technical assistance. This work was supported by Grants-in-Aid for Scientific Research (18H04012 and 21H04785 to M.K., and 20K06862 to H.N.) from the Japan Society for the Promotion of Science (JSPS), Grants-in-Aid for Transformative Research Areas (A) (20H05914 and 20H05915 to M.K., and 20H05916 to M.M.) and AdAMS (16H06276 to K.S.) from the Ministry of Education, Culture, Sports, Science and Technology (MEXT) of Japan.

## Author contributions
H. Nakayama, N. Uesaka, K. Hashimoto, M. Watanabe, and M. Kano designed the study. H Nakayama, T. Miyazaki, Y. Kawamura, M. Choo, K. Konno, and S. Kawata performed the experiments and/or data analyses. M. Abe, M. Yamazaki, and K. Sakimura generated and supplied mutant mice. K. Hashimoto, M. Miyata, M. Watanabe, and M. Kano supervised the study. H. Nakayama, T. Miyazaki, M. Watanabe, and M. Kano wrote the paper.

## Competing interests
The authors declare no competing interests.
