## [Peer review file · Communications Biology]

Reviewers' comments:

Reviewer #1 (Remarks to the Author):

The study submitted by Nakayama et al. is on the role of heterosynaptic interactions in cerebellar Purkinje cells (PC) in the late phase of climbing fiber (CF) elimination. More precisely the role granule cell (GC) to PC and interneuron to PC synapses in the late phase of CF elimination are investigated. Previous reports have pinpointed two key actors involved in this late phase: one is PC type 1 metabotropic glutamate receptor (mGlu1) and its downstream signaling cascade, the other is NMDA-type ionotropic glutamate receptor. The study tries to establish the link between GC activity and each of these putative mechanisms. It convincingly shows that GC activity induces CF elimination through GC to PC transmission-mediated mGlu1 activation. It also supports the conclusion that the NMDA receptor pool which contributes to the overall mechanisms involved in CF elimination is located in molecular layer interneurons. However, the involvement of NMDA receptors in the chain of effects induced by GC activation remains unclear.

Understanding the mechanism enabling synaptic pruning (or elimination) during nervous system development is a key topic and the elimination of cerebellar climbing fiber provides a wonderful model to investigate it. The submitted study follows a long history of studies from the same group on which the authors capitalize to converge onto some robust hypothesis in the submitted paper. By convincingly showing the link between GC activity and CF elimination, mediated through mGlu1 activation, this study will likely remain a historical landmark in the field. The quality of the experiments is excellent. My main concerns are about data interpretation.

Major concern:

(1) After showing that mGlu1 deletion in PC (viral transfection-mediated RNAi-mediated knockdown) results in impaired CF elimination, the authors report an elegant and essential experiment showing that this impairment is occluded by a reduction of the strength of GC inputs on PCs (by knocking out TARP²² in GCs) (Fig. 4j). This neat piece of data is consistent with the idea that GC activity-driven CF elimination is entirely mediated through mGlu1 activation. Yet, the authors further explore the role of NMDA receptor activation as an additional mechanism. They show that partial deletion NMDA receptor functionality in interneurons reduces CF elimination. But they do not provide any data which would enable to conclude that the effect of GC activity is partly mediated by the activation of interneuron's NMDA receptors. Therefore, the part of the study on NMDA role leaves the question of how NMDA receptors are involved in the physiology of CF elimination unsolved.

a. This needs to be discussed specifically: what does support the hypothesis that GC activity-driven interneuron NMDA receptor activation plays a role, given the compelling result shown in Fig. 4j?

b. Given that this part of the study brings little new insight apart from the fact that NMDA receptors born by PC and GC are not involved, I would recommend to move Figs. 6 and 7 to the supplementary material. Doing so, the main finding (Fig. 4j) will highlighted.

(2) The ability of mGlu1 to sense synaptic crosstalk is supported by previous data in the literature (notably Marcaggi and Attwell (2005) and Marcaggi et al. (2009)). Such a property is so relevant to the role of mGluR1 in mediating heterosynaptic interactions that it cannot be ignored. Indeed, both types of synaptic interactions, those mediating synaptic crosstalk between same types of synapses and those which are heterosynaptic, require to be detected. The fact that mGlu1 is key for the detection of them both interestingly suggests that a general function mGlu1 is to detect synaptic interactions. Furthermore, by reporting the activation of a number of nearby GC to PC synapses (Marcaggi & Attwell, 2005), thanks to its kinetic properties enabling preferential sensing of glutamate spillover (Marcaggi et al. 2009), mGlu1 activation provides a reading of the density of GC to PC active synapses in the vicinity of the CF synapse, which is better suited to fine tune the competition between the two types of inputs.

Minor concerns:

Line 70. Please justify the term “elimination” rather than “pruning” and cite a more recent reference about how this mechanism is general.

Line 81. Is it “from” around P7 to P11, or “between” around P7 to P11?

Line 82. Is it “from” around P12 to P17, or “between” around P12 to P17?

Line 102. “This argument (...)” : which argument ? The link with the previous part of the text is unclear.

Fig. 1g. The representation of the “average EPSP” (inset) is irrelevant. It is 12pA in amplitude, which matches the amplitude of the biggest EPSPs visible on the V-clamp trace. So, it cannot be an average EPSP, but an average trace of large EPSPs. Rather, please show a representative not averaged EPSP.

Fig. 2 a-d. In their method to determine the extent of multiple CF innervation, how can the authors ascertain that the first steps in the responses arise from CF synapse activation and not from GC ascending axon inputs? Please comment in the main text.

Line 226. For lobules 1-5, the extent of multiple CF innervation looks different and $P < 0.1$ (Fig. 2d). In such case, the authors cannot conclude that there is no difference. There is no significant difference according to their stats.

Line 260. The authors state that, in $TARP\gamma 2$ -GC-KO, the elimination of CF synapses is impaired in lobules 1-4/5. But the data presented until this line support no change (Fig. 2d; 2i-l), however, data presented later supports a noticeable trend (Fig. 4e). I guess the author meant that translocation of CF was impaired in lobules 1-4/5 (as Fig. 3f suggests)? Please clarify.

Line 393. The authors write “NMDA-mediated EPSCs tended to be smaller, while AMPAR-mediated EPSCs were similar when compared to those of control mice (Fig. 6d, e)”. But Fig. 6e does not show that. It shows that NMDA-mediated EPSCs tended to be smaller, while AMPAR-mediated EPSCs tended to be bigger when compared to those of control mice. Please clarify. Furthermore, the author should analyze the train-evoked charge transfer rather than the peak current and check how it compares.

Lines 380, 386, 389. μM instead of mM .

Line 380. Why using R-CPP and not AP5? Please provide the rationale.

Supplementary Fig. 5b and d: please provide a zoomed y-axis so that NBQX and NBQX + AP5 traces can be compared (as it is, a 50% difference would be within the line thickness).

Reviewer #2 (Remarks to the Author):

The authors examined heterosynaptic effect on developmental climbing fibre (CF) elimination in cerebellar Purkinje cells (PCs) using genetic tools such as various conditional knock-out mice and a lentiviral knock down system, and found heterosynaptic direct pathway (i.e. parallel fibre (PF) activity \rightarrow mGlu1 signaling in PCs \rightarrow CF elimination) and indirect one (i.e. PF activity \rightarrow NMDA receptor activation in molecular layer interneurons (MLIs) \rightarrow MLI-mediated GABAergic inhibition to PCs \rightarrow CF elimination). Their study is well-organized, their argument is straight-forward and clear. Some important points remain to be mentioned and discussed: especially, how mGlu1 in PCs and NMDA receptors in MLIs can be activated during development in a physiological situation and whether PF synapse formation is affected or not in $TARP\gamma 2$ -GC KO mice. As the authors mentioned in the Discussion section of the manuscript, CF innervation can be affected by occupancy competition of PC dendritic spines between PF and CF presynapses. Therefore, the degree of PF synapse formation may affect CF synapse territory and CF elimination, and this possibility should be considered. Some methodological descriptions are not sufficient to

interpret the presented data properly for general readers. Overall, the conclusion of the study is reasonable and convincing. The findings in this study are important, and I believe this study will attract attention from a broad audience in the field of neuroscience.

Major comments

1) Physiological situation of mGlu1 and NMDA receptor activation (i.e. PF burst activity in vivo) during development should be discussed in the manuscript. This study nicely show that PF activity, mGlu1 and NMDA receptor are all essential for CF elimination. As the authors pointed out, activation of both mGlu1 in PCs and NMDA receptors in MLIs requires accumulation of released glutamate in a spatially restricted manner which can be caused by spatially clustered PF burst inputs (Marcaggi, 2015). However, such spatially clustered PF burst inputs during cerebellar development have not reported so far. For example, Fig.1 g, i and j shows that frequency of single GC (i.e, single PF) spontaneous activity in vivo at P12-13 is too low to activate mGlu1 and/or NMDA receptor, and Supplementary Fig 3 shows that there is no obvious mGlu1-mediated slow EPSPs in PCs recorded in vivo at P17. This is a big mystery, and the authors should discuss this point. One of the keys to reconcile this may be that in vivo recording in this study was performed under anesthesia and the real physiological activity might be masked. Moreover, in a really physiological developing situation, patterned spontaneous activity can occur in many developing neural circuits without sensory inputs (Blankenship and Feller, 2010, e.g. retinal waves during development). This may provide some ideas for further intriguing discussion. In the developing cerebellum, Watt et al. (2009) reported patterned travelling waves of spontaneous action potentials among neighboring PCs. It can be reasonably imagined that PFs might also exhibit development-specific spatially-patterned spontaneous activity which would accumulate released glutamate and activate perisynaptic mGlu1 and NMDA receptors.

2) It should be considered and discussed how PF synapse formation is altered in TARPy2-GC KO mice and how it would affect the interpretation of the presented data on CF elimination.

3) The order of the developmental stages of the presented data should be uniform in each figure. In some figures, the data goes from the early to the late stage, but in other figures, the order is reversed. Some figures do not show the developmental stage of the data explicitly. This style of data presentation is confusing and frustrating for general readers. I would suggest using the terminology of developmental stages such as “early”, “mid1”, “mid2”, “late”, and “adult” (or “mature”) and putting the category of the developmental stages in every figure.

Minor comments

4) In Fig. 1 i-k, the authors should specify how to measure the frequency of spontaneous action potential (AP) and EPSC in the Methods section. Was the frequency calculated as the inverse of the interval between the events, or the number of the events divided by recording time? Was the recording time for measuring the frequency long enough to avoid biased estimation?

5) In Fig. 2 e-l2 and Fig. 7 f1-j2, the color of calbindin staining seems gray, not ocher as stated in the figure legend. Immunostaining images should be corrected, and it should be specified what the arrows indicate in the figure legend of Fig. 2.

6) Regarding the data of Fig. 3, the authors should clearly describe how to estimate relative height of VGluT2 terminals and what N/10 μm means. The color code of the staining should be presented. In addition, the relative height of VGluT2 terminals in control seems higher than the one in TARPy2-GC KO mice in the figure. It would be better to draw some lines to clearly indicate the relative height in the figure a2 and b2.

7) Regarding the increase of CF presynaptic terminals in both L8-9 and L1-4/5 of TARPy2-GC KO adult mice, this phenomena should also be discussed in the manuscript. Especially, the increase of CF terminals in L1-4/5 of TARPy2-GC KO adult mice indicates that CF synapse formation can be regulated heterosynaptically (i.e. PF activity to CF synapses) even at the adult stage, although mono CF innervation itself is not altered.

8) In Fig. 5 a and c, the signals in the cerebellum are hard to see. If possible, it would be better to adjust the appearance of the images to make the cerebellar signals visible.

9) In the analysis of freeze-fracture replica electron microscopic images, the authors should explain how to identify PF terminals and MLI dendrites briefly in the Methods section for general readers. I wonder how many mice and cerebellar sections were examined in this analysis and whether the summary of the pooled data can be presented.

Reviewer #3 (Remarks to the Author):

This nice paper provides interesting new insights into synapse elimination at cerebellar climbing fibers during development. It is amazing that after decades of studies, synapse pruning is poorly understood in general and climbing fiber synapse elimination in particular. I have very little criticize about this paper, which I think is eminently suitable for 'communications biology'. My

only question regards Figure 6, where I don't understand why the deletion of GluN1 in molecular layer interneurons does not completely abolish NMDAR EPSCs. GluN1 is an obligatory NMDAR subunit and after its deletion a cell should no longer of NMDAR EPSCs - so why are they still present?

Points of Revision

Responses to reviewers:

We cordially thank the reviewers for their positive evaluation of our work and many constructive suggestions and comments on our manuscript. We have tried our best to address their comments and followed their suggestions as far as possible. We feel that our manuscript has been improved significantly. We have highlighted the changes in the main text and figure legends with light blue markers so that they are easily identifiable.

Responses to Referee #1:

We deeply appreciate Referee #1 for her/his thoughtful and constructive suggestions and comments.

Major concern

(Reviewer's comment)

(1) After showing that mGlu1 deletion in PC (viral transfection-mediated RNAi-mediated knockdown) results in impaired CF elimination, the authors report an elegant and essential experiment showing that this impairment is occluded by a reduction of the strength of GC inputs on PCs (by knocking out TARP γ 2 in GCs) (Fig. 4j). This neat piece of data is consistent with the idea that GC activity-driven CF elimination is entirely mediated through mGlu1 activation. Yet, the authors further explore the role of NMDA receptor activation as an additional mechanism. They show that partial deletion NMDA receptor functionality in interneurons reduces CF elimination. But they do not provide any data which would enable to conclude that the effect of GC activity is partly mediated by the activation of interneuron's NMDA receptors. Therefore, the part of the study on NMDA role leaves the question of how NMDA receptors are involved in the physiology of CF elimination unsolved.

a. This needs to be discussed specifically: what does support the hypothesis that GC activity-driven interneuron NMDA receptor activation plays a role, given the compelling result shown in Fig. 4j?

b. Given that this part of the study brings little new insight apart from the fact that NMDA receptors born by PC and GC are not involved, I would recommend to move

Figs. 6 and 7 to the supplementary material. Doing so, the main finding (Fig. 4j) will highlighted.

(Our response to the comments)

We indeed thank Referee #1 for pointing out this important issue. As for the reviewer's comment (a), we discuss how the activation of interneuron's NMDA receptors presumably driven by GC's activity plays a role in CF synapse elimination (Pages 26-27, Lines 595-620). We followed the reviewer's suggestion (b) and moved the original Figure 6 to Supplemental Fig. 6. We want to keep the original Figure 7 as one of the main figures (new Figure 6) since we strongly feel that the impairment of the late phase of CF elimination in GluN1-MLI/PC-KO mice is an important message in the present study.

(Reviewer's comment)

(2) The ability of mGlu1 to sense synaptic crosstalk is supported by previous data in the literature (notably Marcaggi and Attwell (2005) and Marcaggi et al. (2009)). Such a property is so relevant to the role of mGluR1 in mediating heterosynaptic interactions that it cannot be ignored. Indeed, both types of synaptic interactions, those mediating synaptic crosstalk between same types of synapses and those which are heterosynaptic, require to be detected. The fact that mGlu1 is key for the detection of them both interestingly suggests that a general function mGlu1 is to detect synaptic interactions. Furthermore, by reporting the activation of a number of nearby GC to PC synapses (Marcaggi & Attwell, 2005), thanks to its kinetic properties enabling preferential sensing of glutamate spillover (Marcaggi et al. 2009), mGlu1 activation provides a reading of the density of GC to PC active synapses in the vicinity of the CF synapse, which is better suited to fine tune the competition between the two types of inputs.

(Our response to the comments)

We thank Referee #1 for these comments. We now discuss the ability of mGlu1 to sense synaptic crosstalk and the possibility that such a property may contribute to the heterosynaptic interaction (Pages 24-25, Lines 557-567).

Minor concerns

(Reviewer's comment)

Line 70. Please justify the term “elimination” rather than “pruning” and cite a more recent reference about how this mechanism is general.

(Our response to the comments)

We changed the term “elimination” to “pruning” (Page 4, Line 73). We have cited a review article (ref 6) that shows the postnatal development of CF-PC synapse is an example of activity-dependent synaptic pruning.

(Reviewer’s comment)

Line 81. Is it “from” around P7 to P11, or “between” around P7 to P11?

(Our response to the comments)

It is “from” around P7 to around P11” (Page 4, Line 82).

(Reviewer’s comment)

Line 82. Is it “from” around P12 to P17, or “between” around P12 to P17?

(Our response to the comments)

It is “from” around P12 to around P17” (Page 4, Line 83).

(Reviewer’s comment)

Line 102. “This argument (...)” : which argument ? The link with the previous part of the text is unclear.

(Our response to the comments)

We apologize for the unclear description. We have added the clause to explain which argument it is.

“This argument that mGlu1 is activated at PF-PC synapses is...” (Page 5, Line 103).

(Reviewer's comment)

Fig. 1g. The representation of the “average EPSP” (inset) is irrelevant. It is 12pA in amplitude, which matches the amplitude of the biggest EPSPs visible on the V-clamp trace. So, it cannot be an average EPSP, but an average trace of large EPSPs. Rather, please show a representative not averaged EPSP.

(Our response to the comments)

We thank Referee #1 for this comment.

The inset shows an average EPSC, not an average EPSP. We checked the original data and found that the scale bars for the lower traces (EPSCs), and the inset (average EPSC) were incorrect. The correct values are 5 pA and 100 ms (lower traces, EPSCs), and 2 pA and 10 ms (inset, average EPSC). We corrected the legend for Figure 1g and 1h accordingly (Page 46, Line 1121).

(Reviewer's comment)

Fig. 2 a-d. In their method to determine the extent of multiple CF innervation, how can the authors ascertain that the first steps in the responses arise from CF synapse activation and not from GC ascending axon inputs? Please comment in the main text.

(Our response to the comments)

We thank Referee #1 for pointing out this important issue.

We now explain how we judged that the first steps of EPSCs were elicited by stimulation of CFs, not by activation of GC ascending axons (Page 10, Lines 218-224).

(Reviewer's comment)

Line 226. For lobules 1-5, the extent of multiple CF innervation looks different and P is < 0.1 (Fig. 2d). In such case, the authors cannot conclude that there is no difference. There is no significant difference according to their stats.

(Our response to the comments)

We accepted the reviewer's advice and amended the sentence accordingly (Page 11, Line 235).

(Reviewer's comment)

Line 260. The authors state that, in TARPγ2-GC-KO, the elimination of CF synapses is impaired in lobules 1-4/5. But the data presented until this line support no change (Fig. 2d; 2i-l), however, data presented later supports a noticeable trend (Fig. 4e). I guess the author meant that translocation of CF was impaired in lobules 1-4/5 (as Fig. 3f suggests)? Please clarify.

(Our response to the comments)

Our electrophysiological data for the number of CF EPSC-steps show a trend of slight impairment of CF synapse elimination in lobules 1-4/5 of TARPγ2-GC-KO mice (Fig. 2d and Fig. 4f (not "Fig. 4e") but the trend did not reach the statistically significant level in either case ($P = 0.097$. for Fig. 2d; $P = 0.081$ for Fig. 4f). On the other hand, our morphological data show that the number of VGluT2-positive CF synapses remaining on the soma in lobules 1-4/5 was significantly higher ($P = 0.013$) in TARPγ2-GC-KO mice than in control mice at 8 weeks of age (Fig. 3f). Therefore, we state that "...CF synapse elimination was slightly impaired in the anterior lobules of TARPγ2-GC-KO mice" (Page 12, Lines 272-273).

(Reviewer's comment)

Line 393. The authors write "NMDA-mediated EPSCs tended to be smaller, while AMPAR-mediated EPSCs were similar when compared to those of control mice (Fig. 6d, e)". But Fig. 6e does not show that. It shows that NMDA-mediated EPSCs tended to be smaller, while AMPAR-mediated EPSCs tended to be bigger when compared to those of control mice. Please clarify. Furthermore, the author should analyze the train-evoked charge transfer rather than the peak current and check how it compares.

(Our response to the comments)

We thank Referee #1 for these comments.

We followed Referee #1's suggestion and analyzed the train-evoked charge transfer of the AMPA receptor-mediated component and the NMDA receptor-mediated component. We now show both the data for the peak current and those for the charge transfer in the new Supplementary Figure 6. We found that the AMPAR-mediated EPSC charge transfer was similar between control and GluN1-MLI/PC-KO mice (new Supplementary Fig. 6i), although the peak amplitude of AMPAR-EPSCs tended to be bigger in GluN1-MLI/PC-KO mice than in control mice (new Supplementary Fig. 6f).

Therefore, we keep our original statement that "NMDA-mediated EPSCs tended to be smaller, while AMPAR-mediated EPSCs were similar when compared to those of control mice" (Page 19, Lines 443-445).

(Reviewer's comment)

Lines 380, 386, 389. μ M instead of mM.

(Our response to the comments)

We have amended the mistakes (Page 19, Lines 424, 426, 432).

(Reviewer's comment)

Line 380. Why using R-CPP and not AP5? Please provide the rationale.

Supplementary Fig. 5b and d: please provide a zoomed y-axis so that NBQX and NBQX + AP5 traces can be compared (as it is, a 50% difference would be within the line thickness).

(Our response to the comments)

We thank the reviewer for these comments. To be consistent between the data for MLIs and PCs in terms of evaluating NMDA receptor-mediated components in synaptic responses, we performed additional experiments to examine the effect of R-CPP on CF-EPSCs and PF-EPSCs in control mice. We obtained essentially the same results for CF-EPSCs as we did by using AP5. On the other hand, our reexamination of tetanus-induced PF EPSCs shows that they

contained NMDA receptor-mediated components. We now replaced the previous data using AP5 with the new data using α -CPP (new Supplemental Fig. 7). Moreover, we quantified the NMDA receptor-mediated component of CF-EPSCs and tetanus-induced PF-EPSCs at P15-17 and 7 weeks of age by measuring the charge transfer (Supplementary Fig. 7e-f).

Responses to Reviewer #2:

We appreciate reviewer #2's positive evaluation of our work and thank her/him for the constructive suggestions and comments. Our point-by-point responses to the comments are as follows.

Major comments

(Reviewer's comment)

1) Physiological situation of mGlu1 and NMDA receptor activation (i.e. PF burst activity in vivo) during development should be discussed in the manuscript. This study nicely show that PF activity, mGlu1 and NMDA receptor are all essential for CF elimination. As the authors pointed out, activation of both mGlu1 in PCs and NMDA receptors in MLIs requires accumulation of released glutamate in a spatially restricted manner which can be caused by spatially clustered PF burst inputs (Marcaggi, 2015). However, such spatially clustered PF burst inputs during cerebellar development have not reported so far. For example, Fig.1 g, i and j shows that frequency of single GC (i.e. single PF) spontaneous activity in vivo at P12-13 is too low to activate mGlu1 and/or NMDA receptor, and Supplementary Fig 3 shows that there is no obvious mGlu1-mediated slow EPSPs in PCs recorded in vivo at P17. This is a big mystery, and the authors should discuss this point. One of the keys to reconcile this may be that in vivo recording in this study was performed under anesthesia and the real physiological activity might be masked. Moreover, in a really physiological developing situation, patterned spontaneous activity can occur in many developing neural circuits without sensory inputs (Blankenship and Feller, 2010, e.g. retinal waves during development). This may provide some ideas for further intriguing discussion. In the developing cerebellum, Watt et al. (2009) reported patterned travelling waves of spontaneous action potentials among neighboring PCs. It can be reasonably imagined that PFs might also exhibit development-specific spatially-patterned spontaneous activity which

would accumulate released glutamate and activate perisynaptic mGlu1 and NMDA receptors.

(Our response to the comments)

We indeed thank Referee #2 for pointing out these important issues. According to the suggestions raised by the reviewer, we now discuss several possibilities for how mGlu1 and NMDA receptors can be activated *in vivo* by clustered PF burst inputs (Pages 27-28, Lines 621-641).

(Reviewer's comment)

2) It should be considered and discussed how PF synapse formation is altered in TARPy2-GC KO mice and how it would affect the interpretation of the presented data on CF elimination.

(Our response to the comments)

We thank Referee #2 for this constructive comment and for giving us the chance to examine PF-PC synapses in TARPy2-GC KO mice. We examined the morphology of PF-PC synapses quantitatively using electron microscopy and the stimulus-response relationship of PF-EPSCs (new Supplementary Figure 4; Pages 12-14, Lines 274-303). The results show no significant differences between control and TARPy2-GC KO mice in the number of PF-PC synapses, the PF terminal area, or the stimulus-response relationship of PF-EPSCs.

(Reviewer's comment)

3) The order of the developmental stages of the presented data should be uniform in each figure. In some figures, the data goes from the early to the late stage, but in other figures, the order is reversed. Some figures do not show the developmental stage of the data explicitly. This style of data presentation is confusing and frustrating for general readers. I would suggest using the terminology of developmental stages such as "early", "mid1", "mid2", "late", and "adult" (or "mature") and putting the category of the developmental stages in every figure.

(Our response to the comments)

We thank Referee #2 for the advice to improve our presentation.

We are sorry that our style of presentation caused a confusing and frustrating impression. Since the CF mono innervation pattern is established by the end of the third postnatal week, we first examined the CF innervation pattern by counting the number of CF-EPSC steps in mice older than P21 to check whether CF synapse elimination occurred normally or not. If we found persistent multiple CF innervation, then we examined the CF innervation during the initial three postnatal weeks to determine at what developmental stage the impairment of CF synapse elimination emerges. To improve our presentation, we followed the reviewer's suggestion and put the stage of mouse development (e.g., "Adolescent", "Adult") or the stage of CF synapse elimination (e.g., "Early phase", "Late phase") in each figure panel.

Minor comments

(Reviewer's comment)

4) In Fig. 1 i-k, the authors should specify how to measure the frequency of spontaneous action potential (AP) and EPSC in the Methods section. Was the frequency calculated as the inverse of the interval between the events, or the number of the events divided by recording time? Was the recording time for measuring the frequency long enough to avoid biased estimation?

(Our response to the comments)

We calculated the frequencies of spontaneous action potentials (APs) and EPSCs in PCs and GCs as the number of events divided by a recording time of 30 seconds when the baseline of membrane current or potential became stable enough after establishing whole-cell recordings. We describe these points in the Methods section (Page 32, Lines 739-743).

(Reviewer's comment)

5) In Fig. 2 e-l2 and Fig. 7 f1-j2, the color of calbindin staining seems gray, not ocher as stated in the figure legend. Immunostaining images should be corrected, and it should be specified what the arrows indicate in the figure legend of Fig. 2.

(Our response to the comments)

We have corrected the legends for Fig. 2 and Fig. 6 from (blue or ocher) to (blue or gray) (Page 47, Line 1140; Page 55, Line 1229). We added the following sentence to the Fig. 2 legend to explain what the arrows indicate. "Arrows in h1 and h2 indicate anterogradely unlabeled VGluT2-positive terminals on PCs." (Page 48, Lines 1145-1146)

(Reviewer's comment)

6) Regarding the data of Fig. 3, the authors should clearly describe how to estimate relative height of VGluT2 terminals and what N/10 μm means. The color code of the staining should be presented. In addition, the relative height of VGluT2 terminals in control seems higher than the one in TARP γ 2-GC KO mice in the figure. It would be better to draw some lines to clearly indicate the relative height in the figure a2 and b2.

(Our response to the comments)

We thank Referee #2 for the valuable comments and suggestions. In the Fig. 3 legend, we explained how we estimated the relative height of VGluT2 terminals (Page 49, Lines 1160-1162) and what N/10 μm means (Page 49, Lines 1165-1166) and described the color codes of the immunostaining (Page 49, Line 1153). We indicated the pial surface with a dotted line in each of Fig. 3a1-d2.

(Reviewer's comment)

7) Regarding the increase of CF presynaptic terminals in both L8-9 and L1-4/5 of TARP γ 2-GC KO adult mice, this phenomena should also be discussed in the manuscript. Especially, the increase of CF terminals in L1-4/5 of TARP γ 2-GC KO adult mice indicates that CF synapse formation can be regulated heterosynaptically (i.e. PF

activity to CF synapses) even at the adult stage, although mono CF innervation itself is not altered.

(Our response to the comments)

We thank Referee #2 for raising an important issue.

We now discuss how CF translocation was enhanced whereas PF-PC synapses were normal in lobules 8-9 of TARPy2-GC KO mice (Pages 23-24, Lines 520-539).

(Reviewer's comment)

8) In Fig. 5 a and c, the signals in the cerebellum are hard to see. If possible, it would be better to adjust the appearance of the images to make the cerebellar signals visible.

(Our response to the comments)

We thank Referee #2 for this suggestion.

In the figure panels showing GluN1 immuno-staining, we have added the panels that show part of the cerebellum with higher magnification (new Figure 5b and e).

(Reviewer's comment)

9) In the analysis of freeze-fracture replica electron microscopic images, the authors should explain how to identify PF terminals and MLI dendrites briefly in the Methods section for general readers. I wonder how many mice and cerebellar sections were examined in this analysis and whether the summary of the pooled data can be presented.

(Our response to the comments)

We thank Referee #2 for the comment and suggestion.

We now describe how to identify PF terminals and MLI dendrites briefly in the Methods section (Page 34, Line 786-791). We now present the summary data on the density of GluN1 immuno-particles with the numbers of mice and cerebellar sections (new Supplementary Fig. 6c; Page 18, Lines 417-419).

Responses to Reviewer #3:

We appreciate reviewer #3's positive evaluation of our work.

(Reviewer's comment)

This nice paper provides interesting new insights into synapse elimination at cerebellar climbing fibers during development. It is amazing that after decades of studies, synapse pruning is poorly understood in general and climbing fiber synapse elimination in particular. I have very little criticize about this paper, which I think is eminently suitable for 'communications biology'. My only question regards Figure 6, where I don't understand why the deletion of GluN1 in molecular layer interneurons does not completely abolish NMDAR EPSCs. GluN1 is an obligatory NMDAR subunit and after its deletion a cell should no longer of NMDAR EPSCs - so why are they still present?

(Our response to the comments)

We thank Referee #3 for the question. As the referee pointed out, NMDA receptor-mediated component of PF-induced EPSC in MLIs was not completely deleted in GluN1/MLI-KO mice. Indeed, after scrutinizing the individual data about NMDA receptor-mediated component of PF-EPSCs in MLIs, we noticed that among 8 MLIs of NR1-MLI/PC-KO mice, 3 cells had almost no NMDAR-mediated component, whereas 5 cells exhibited clear NMDAR-components. We now mention this point in the Discussion (Page 27, Lines 610-616).

REVIEWERS' COMMENTS:

Reviewer #1 (Remarks to the Author):

The authors have addressed my concerns. The MS is suitable for publication. The only point I would stress is that the last paragraph of the Discussion is a bit pushy. I do not think that one can say that the question of "how PF inputs heterosynaptically influence CF synapse elimination" is fully "elucidated".

Reviewer #2 (Remarks to the Author):

The authors have revised the manuscript reasonably and all the previous concerns have been addressed appropriately. In Figs 2 and 6, and their legends, there are still some typos("l, j1, j2" must be "i, j1, j2" and "e, g, l, k" must be "e, g, i, k", and "ff2" should be "f2" in Fig 2) and ambiguous presentation (the color code "cyan" should be explained explicitly for clarity in Fig.2 f1, h1 and Fig.6 h1 and j1). However, they are not critical and I think it is up to the authors to correct them. The manuscript has been greatly improved and I recommend its publication.

Reviewer #3 (Remarks to the Author):

Accept - no more comments

Points of Revision

Responses to reviewers:

We cordially thank the reviewers for their positive evaluation of our work and careful manuscript reading. We have highlighted the changes in the main text in red so that they are easily identifiable.

Responses to Referee #1:

We deeply appreciate Referee #1 for her/his constructive suggestion.

(Reviewer's comment)

The authors have addressed my concerns. The MS is suitable for publication. The only point I would stress is that the last paragraph of the Discussion is a bit pushy. I do not think that one can say that the question of "how PF inputs heterosynaptically influence CF synapse elimination" is fully "elucidated"..

(Our response to the comments)

We toned down our statement as follows. We changed the wording of the first sentence of the final paragraph of the Discussion (Page 28, line 641-642) as follows:

"In conclusion, our present results strongly suggest that PF inputs heterosynaptically influence CF synapse elimination in the developing cerebellum."

Responses to Referee #2:

We thank Referee #2's careful reading of our manuscript.

(Reviewer's comment)

The authors have revised the manuscript reasonably and all the previous concerns have been addressed appropriately. In Figs 2 and 6, and their legends, there are still some typos ("l, j1, j2" must be "i, j1, j2" and "e, g, l, k" must be "e, g, i, k", and "ff2" should be "f2" in Fig 2) and ambiguous presentation (the color code "cyan" should be

explained explicitly for clarity in Fig.2 f1, h1 and Fig.6 h1 and j1). However, they are not critical and I think it is up to the authors to correct them. The manuscript has been greatly improved and I recommend its publication.

(Our response to the comments)

We have corrected the mistakes and typos in Figure 2 legend. We added the explanation about the anterogradely unlabeled (DA488-negative) VGluT2-positive CF terminals (arrows) in the legends of Fig. 2 (page 46, lines 1144-1145) and Fig. 6 (page 50, lines 1229-1230).